# Turning Normalizing Flows into Monge Maps with Geodesic Gaussian Preserving Flows

**Guillaume Morel**                                                        *guillaume.morel@imt-atlantique.fr*
*IMT Atlantique, LaTIM, U1101, Brest, France*

**Lucas Drumetz**                                                          *lucas.drumetz@imt-atlantique.fr*
*IMT Atlantique, Lab-STICC, UMR CNRS 6285, Brest, France*

**Simon Benaichouche**                                                     *simon.benaichouche@imt-atlantique.fr*
*IMT Atlantique, Lab-STICC, UMR CNRS 6285, Brest, France*

**Nicolas Courty**                                                         *nicolas.courty@irisa.fr*
*Université Bretagne sud, IRISA, UMR CNRS 6074, Vannes, France.*

**François Rousseau**                                                      *francois.rousseau@imt-atlantique.fr*
*IMT Atlantique, LaTIM, U1101, Brest, France*

**Reviewed on OpenReview:** *https://openreview.net/forum?id=2UQv8L1Cv9*

## Abstract

Normalizing Flows (NF) are powerful likelihood-based generative models that are able to trade off between expressivity and tractability to model complex densities. A now well established research avenue leverages optimal transport (OT) and looks for Monge maps, i.e. models with minimal effort between the source and target distributions. This paper introduces a method based on Brenier's polar factorization theorem to transform any trained NF into a more OT-efficient version without changing the final density. We do so by learning a rearrangement of the source (Gaussian) distribution that minimizes the OT cost between the source and the final density. The Gaussian preserving transformation is implemented with the construction of high dimensional divergence free functions and the path leading to the estimated Monge map is further constrained to lie on a geodesic in the space of volume-preserving diffeomorphisms thanks to Euler's equations. The proposed method leads to smooth flows with reduced OT costs for several existing models without affecting the model performance. The code is available here `https://github.com/morel-g/GPFlow`.

## 1 Introduction

Modeling high dimensional data is a central question in data science as they are ubiquitous in applications. Various tasks such as probabilistic inference, density estimation or sampling of new data require accurate probabilistic models that need to be defined efficiently. There exists a large variety of generative models in the literature. Among other approaches, diffusion / score based models (Ho et al., 2020; Sohl-Dickstein et al., 2015; Song & Ermon, 2019), variational autoencoders (VAES) (Kingma & Welling, 2014; Rezende et al., 2014) and generative adversarial networks (GAN) (Goodfellow et al., 2014) are frequent choices, each with their strengths and weaknesses.

**Normalizing flows.** A fourth popular class of generative models is Normalizing flows (NF). NF models transform a known probability distribution (Gaussian in most cases) into a complex one allowing for efficient sampling and density estimation. To do so they use a diffeomorphism $\mathbf{f} : \mathbb{R}^d \to \mathbb{R}^d$ which maps a target probability distribution $\mu$ to the known source distribution $\nu = \mathbf{f}_{\#}\mu$ (Dinh et al., 2014; Rezende & Mohamed, 2015). In practice the flow satisfies the change of variables formula:

$$\log p_\mu(\mathbf{x}) = \log p_\nu(\mathbf{f}(\mathbf{x})) + \log|\det \nabla \mathbf{f}(\mathbf{x})|. \tag{1}$$

There are many possible parameterizations of $\mathbf{f}$, usually relying on automatic differentiation to train their parameters via first order optimization algorithms. For density estimation applications, training is done by maximizing the likelihood of the observed data. The data are generally high dimensional and accessing $p_\mu(\mathbf{x})$ for a given $\mathbf{x}$ requires computing determinant of the Jacobian matrix of $\mathbf{f}$. This operation has a complexity of $O(d^3)$ in general and thus a requirement for the flow architecture is to have a tractable determinant of the Jacobian while remaining expressive enough (Dinh et al., 2014; Kingma & Dhariwal, 2018; Rezende & Mohamed, 2015; Papamakarios et al., 2021).

**Optimal transport.** A diffeomorphism transforming any well-behaved distribution into another always exists in theory (Papamakarios et al., 2021). However, there can be many ways to transform one probability measure $\mu$ into another probability measure $\nu$, and therefore the function $\mathbf{f}$ is generally not unique. This has led to many proposed architectures in the literature (Kingma & Dhariwal, 2018; Grathwohl et al., 2018; Huang et al., 2018; De Cao et al., 2020; Papamakarios et al., 2017). The question of choosing the "best" transformation among all existing ones is therefore crucial, independently of how accurately $\mu$ models the target distribution. One way to make the architecture unique (under appropriate conditions on the two distributions) is to use optimal transport (Hamfeldt, 2019; Peyré et al., 2017; Santambrogio, 2015; Villani, 2008), that is to choose the one giving the Wasserstein distance between $\mu$ and $\nu$, with a squared $\mathcal{L}_2$ ground cost:

$$W_2^2(\mu, \nu) = \min_{\mathbf{f}} \int_{\mathbb{R}^d} |\mathbf{f}(\mathbf{x}) - \mathbf{x}|^2 d\mu(\mathbf{x}), \quad \nu = \mathbf{f}_{\#}\mu, \tag{2}$$

where $\mathbf{f}_{\#}\mu$ denotes the push forward operator of $\mu$ through $\mathbf{f}$. An optimal model in the sense of (2) minimizes the total mass displacement which can be a desirable property even if it is often a difficult task. In particular Brenier's theorem (Brenier, 1991) states that the optimal function $\mathbf{f}$ is the gradient of a scalar convex function, which is widely used when solving (2).

One key property of OT mappings is that they should better preserve the structure of the distribution compared to non OT transformations. This makes them particularly appealing for machine learning applications, and may also help with generalization performance (Karkar et al., 2020).

**Optimal transport and NF models.** Including OT in NF models has recently received much attention with various approaches to obtain a map $\mathbf{g}$ which satisfies the property (2). Among all these methods, many use either directly Brenier's theorem (Brenier, 1991) or the dynamic OT formulation with the Benamou-Brenier approach (Benamou & Brenier, 2000). One important remark is that most of the approaches considered need dedicated architectures in order to satisfy the OT property. For example the transformation $\mathbf{f}$ is often written as neural network modeling the gradient of a (possibly convex) scalar function (Amos et al., 2016; Finlay et al., 2020a; Huang et al., 2020; Onken et al., 2020; Zhang et al., 2018). This sometimes requires some particular training process (Finlay et al., 2020a; Huang et al., 2020; Onken et al., 2020) and/or the addition of some penalization terms in the loss function (Onken et al., 2020; Finlay et al., 2020b; Yang & Karniadakis, 2020). When considering the Benamou-Brenier formulation, the normalizing flow is interpreted as the discretization of a continuous ordinary differential equation (Chen et al., 2018b) and the optimal transport problem is then solved dynamically (Finlay et al., 2020b; Onken et al., 2020; Zhang et al., 2018).

The methods mentioned above require to constrain the architecture and/or the training procedure which can be problematic in some cases and has already motivated other OT related approaches (Uscidda & Cuturi, 2023). In this paper we present a method which achieve the optimal map of a given NF without constraining the architecture or the training procedure.

## 1.1 Main contributions

**Polar factorization.** An overlooked implication of Brenier's theorem is the so-called polar factorization theorem, that states that the optimal transport map $\nabla\psi$ solving (2) can be factorized into the composition of two functions $\nabla\psi = \mathbf{s} \circ \mathbf{f}$, the function $\mathbf{f}$ being some arbitrary smooth map from $\mu$ to $\nu$ and $\mathbf{s}$ an associated measure preserving function of $\nu$ (Brenier, 1991). The idea we exploit is the possibility, from a given flow $\mathbf{f}$ (and its corresponding inverse $\mathbf{g} = \mathbf{f}^{-1}$), to rearrange the distribution $\nu$ using $\mathbf{s}$ to obtain a new map reducing the OT cost without changing the distribution given by the push-forward $\mu = \mathbf{g}_{\#}\nu = (\mathbf{g} \circ \mathbf{s}^{-1})_{\#}\nu$. The OT-improved map can then be obtained with the composition $\mathbf{g} \circ \mathbf{s}^{-1}$. Interestingly

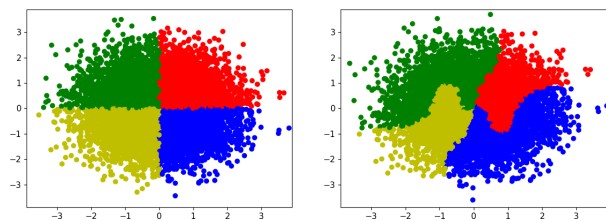

Figure 1: A GP transformation applied on particles sampled from a two-dimensional Gaussian distribution. The mean and standard deviation stay the same; only the positions of the particles change.

this property is not as popular as the previous one and to our knowledge is not used when dealing with OT and NF models. Yet normalizing flows can take advantage of this formulation mostly because the distribution $\nu$ is known and simple (here and in the following $\nu$ is a standard normal) which makes it possible to construct architectures preserving $\nu$.

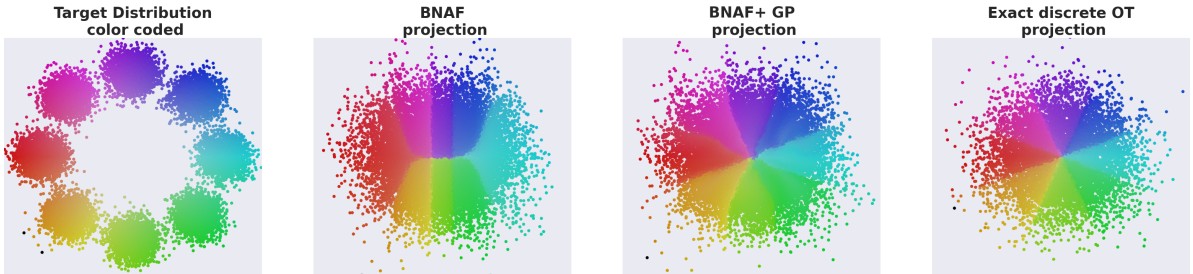

Figure 2: Eight Gaussians test case with colored distributions. A GP flow is trained on a pre-trained BNAF model (De Cao et al., 2020) to reduce the OT cost.

**Gaussian preserving flows.** Our work differs from the state of the art as we do not propose a new normalizing flow model. Instead, we propose to use Brenier's theorem to compute the Monge map for any pre-existing architecture. Indeed there exists a wide variety of architecture available in the literature (Kingma & Dhariwal, 2018; Grathwohl et al., 2018; Huang et al., 2018; De Cao et al., 2020; Papamakarios et al., 2017) each with their pros and cons which sometimes depend specifically on the test case considered. Our idea is to use Brenier's polar factorization theorem to rearrange the points in the known distribution to obtain the optimal map associated with a given flow. We consider the most common case where the known distribution is a standard normal and call such rearranging maps Gaussian Preserving (GP) flows, see Figure 1. An important point is that by construction our GP map will only change the OT cost of the model. The target density and therefore the training loss given by the model will stay the same. This allows us to take any pre-trained model and compute the associated Monge map, thus improving the model in terms of OT displacement from the source to the target distribution, without changing the modeled density see Figure 2.

**Construction of divergence free functions in high dimensions.** We show that divergence free functions can be used to model GP flows. We therefore derive an effective construction of divergence free functions in high dimensions and apply it in the latent space of some popular VAE models.

**Euler's equations.** Since several GP flow models can solve the same OT problem, we also look for a way to find the "best" GP flow. This is somehow similar to the approaches from Finlay et al. (2020b); Onken et al. (2020) where the trajectories of a continuous normalizing flow are penalized to be straight lines. This is not strictly needed to find the Monge map but can be interpreted as some geodesic over all the flows which solve the associated OT problem. In this work, we show that the geodesics associated with the OT problem are actually given by solutions to the Euler equations, following a celebrated result by Arnold (Arnold, 1966).

The penalization of Euler's equations in high dimensions and its practical implementation is therefore also considered, which is to the best of our knowledge an original contribution.

**Data structure preservation with optimal transport.** Finally we show one potential interest of GP flows by studying the preservation of the data structure experimentally. More specifically we focus on the preservation of disentanglement on the dSprites (Matthey et al., 2017), MNIST (Lecun et al., 1998) and Chairs (Mathieu et al., 2014) datasets in some variational auto-encoder (VAE) latent space. On this particular example we show that OT allows to improve the preservation of the structure of the latent data points which is otherwise destroyed when applying the NF model.

## 2 Polar factorization theorem

The main idea is to use the Brenier's polar factorization theorem to construct the Monge map with a rearrangement of the known probability distribution $\nu$.

**Notation and measure preserving definition.** In the following we will use the shortened notation $(\mathcal{X}, \nu)$ for a probability space $(\mathcal{X}, \mathcal{B}, \nu)$ where $\mathcal{X}$ is a set, $\mathcal{B}$ a $\sigma$-algebra of its subset and $\nu$ a probability distribution. We recall the definition of measure preserving function use in Brenier (1991): a measure preserving mapping from a probability space $(\mathcal{X}, \nu)$ into itself is a mapping $\mathbf{s} : \mathcal{X} \to \mathcal{X}$ such that for every $\nu$ measurable subset $A$ of $\mathcal{X}$, $\mathbf{s}^{-1}(A)$ is $\nu$-measurable and $\nu(\mathbf{s}^{-1}(A)) = \nu(A)$.

**Theorem 1** (Brenier's polar factorization (Brenier, 1991)). *Let $(\mathcal{X}, \nu)$ be a probability space, $\mathcal{X}$ bounded. Then for each $\mathbf{g} \in L^p(\mathcal{X}, \nu, \mathbb{R}^d)$ satisfying the non degeneracy condition*

$$\nu(\mathbf{g}^{-1}(E)) = 0 \text{ for each Lebesgue negligible subset } E \text{ of } \mathbb{R}^d,$$

*there exists a unique convex function $\psi : \mathcal{X} \to \mathbb{R}$ and a unique measure preserving function $\mathbf{s} : \mathcal{X} \to \mathcal{X}$ such that*

$$\mathbf{g}(\mathbf{s}(\mathbf{x})) = \nabla \psi(\mathbf{x}),$$

*and $\mathbf{s}(\mathbf{x})$ minimizes the cost $\int_{\mathcal{X}} |\mathbf{g}(\mathbf{s}(\mathbf{x})) - \mathbf{x}|^2 d\nu(\mathbf{x})$.*

Note that in his work Brenier mention the maximization of $\int \mathbf{x} \cdot \mathbf{g} \circ \mathbf{s}(\mathbf{x}) d\nu(\mathbf{x})$ which is equivalent to the minimization with repect to $\mathbf{s}$ of the quadratic cost thanks to the measure preserving property of $\mathbf{s}$. Our goal is to leverage the polar factorization theorem in order to solve the OT problem between $\nu$ and $\mu := \mathbf{g}_{\#}\nu$ where $\mathbf{g}$ is given and $\nu = \mathcal{N}(\mathbf{0}, \mathrm{Id})$, by looking for the rearrangement $\mathbf{s}$ via an optimization problem. To do so we need to construct a class of measure preserving maps.

**Remark 1.** *Since in practice we consider $\nu$ to be a standard normal, the domain $\mathcal{X}$ is not bounded and therefore does not strictly satisfy the hypothesis of Theorem 1. We do not investigate this point further and simply quote a remark from Brenier's work (Brenier, 1991): "we believe that the result is still true when $\mathcal{X}$ is unbounded, provided that $p > 1$ and $\int_{\mathcal{X}} \|\mathbf{x}\|^q \beta(\mathbf{x}) d\mathbf{x} < +\infty$, where $1/q + 1/p = 1$". The function $\beta(\mathbf{x}) = e^{-\|\mathbf{x}\|^2/2}$ is the probability density of $\nu$, and the inequality is therefore satisfied.*

## 3 Gaussian preserving flows

In order to apply Brenier's polar factorization theorem, it is therefore needed to construct a class of measure preserving maps. Since we consider the case where $\nu$ is a standard normal, we call such maps Gaussian preserving (GP). All proofs of the propositions and lemmas are given in Appendix B.

Consider two probability measures $\alpha$ and $\beta$ with density $h_\alpha$ and $h_\beta$ respectively. A map $\mathbf{s}$ is measure preserving between $\alpha$ and $\beta$ if it satisfies the change of variable equality (same as (1) without the log) $h_\alpha(\mathbf{x}) = h_\beta(\mathbf{s}(\mathbf{x}))|\det(\nabla\mathbf{s}(\mathbf{x}))|$. In our case, we want $\mathbf{s}$ to be Gaussian preserving therefore $h_\alpha = h_\beta = e^{-\|\mathbf{x}\|^2/2}$ and one gets

$$|\det \nabla\mathbf{s}(\mathbf{x})| = e^{(\|\mathbf{s}(\mathbf{x})\|^2 - \|\mathbf{x}\|^2)/2}. \tag{3}$$

It turns out that Lebesgue preserving functions (i.e. satisfying $|\det \nabla\phi| = 1$) can be used to construct maps satisfying (3). In the following we will denote $\mathbf{erf} : \mathbb{R}^d \to \mathbb{R}^d$ the distribution function of a one dimensional Gaussian (that is $\mathrm{erf}(x) = \frac{2}{\sqrt{\pi}} \int_0^x e^{-t^2} dt$) applied component wise.

**Proposition 1.** *Let $\Omega = (-1, 1)^d$. The map $\mathbf{s}$ is a smooth Gaussian preserving function (i.e. satisfying (3)) if and only if there exists $\boldsymbol{\phi} : \Omega \to \Omega$ such that $|\det \nabla \boldsymbol{\phi}| = 1$ and*

$$\mathbf{s}(\mathbf{x}) = \sqrt{2}\,\mathbf{erf}^{-1} \circ \boldsymbol{\phi} \circ \mathbf{erf}(\frac{\mathbf{x}}{\sqrt{2}}), \quad \mathbf{x} \in \mathbb{R}^d.$$

From now on we will focus on the construction of volume and orientation preserving maps (i.e. satisfying $\det \nabla \boldsymbol{\phi} = 1$) since functions satisfying $\det \nabla \boldsymbol{\phi} = -1$ can be constructed from them see Appendix B.1.2. Moreover, one has the following result regarding the regularity of GP flows.

**Lemma 1.** *Assume the Monge map and the NF architecture are $C^1$ diffeomorphisms. Then the corresponding GP flow $\mathbf{s}$ is $C^1$, the associated function $\boldsymbol{\phi}$ is also $C^1$ and either satisfies $\det \nabla \boldsymbol{\phi}(\mathbf{x}) = 1$ everywhere or $\det \nabla \boldsymbol{\phi}(\mathbf{x}) = -1$ everywhere.*

### 3.1 Volume-orientation preserving maps

First we introduce the space $\mathrm{SDiff}(\Omega)$ we will working with from now on. Let $\mathrm{Diff}(\Omega)$ be the set of all diffeomorphisms in $\Omega$ then $\mathrm{SDiff}(\Omega) := \{\boldsymbol{\psi} \in \mathrm{Diff}(\Omega) \mid \det(\nabla \boldsymbol{\psi})(\mathbf{x}) = 1, \ \forall \mathbf{x} \in \Omega\}$, where $\Omega = (-1, 1)^d$. That is we need a transformation which satisfies two properties: 1) the function must be volume and orientation preserving, 2) the solution must stay in the domain $(-1, 1)^d$. Consider the following ODE:

$$\begin{cases} \frac{d}{dt}\mathbf{X}(t, \mathbf{x}) = \mathbf{v}(t, \mathbf{X}(t, \mathbf{x})), & \mathbf{x} \in \Omega, \quad 0 \le t \le T, \\ \mathbf{X}(0, \mathbf{x}) = \mathbf{x}. \end{cases} \tag{4}$$

We impose two conditions on the velocity $\mathbf{v}$:

$$\nabla \cdot \mathbf{v} = 0, \quad \text{in } \Omega, \tag{5}$$

$$\mathbf{v} \cdot \mathbf{n} = 0, \quad \text{on } \partial\Omega, \tag{6}$$

where $\mathbf{n}$ is the outward normal at the boundary of $\Omega$. We define $\boldsymbol{\phi}$ to be the solution at the final time $\boldsymbol{\phi}(\mathbf{x}) := \mathbf{X}(T, \mathbf{x})$. Property (5) implies that $\det \nabla \boldsymbol{\phi} = 1$ (this can be checked with the formula $\frac{d}{dt} \det \nabla \mathbf{X} = \mathrm{div}(\mathbf{v}) \det \nabla \mathbf{X}$), and property (6) ensures that $\boldsymbol{\phi}$ has values in $\Omega$. Any function in $\mathrm{SDiff}(\Omega)$ can be written as a solution to (4) for $d \ge 3$ (Shnirelman, 1993), for $d = 2$ some pathological cases can be constructed (Shnirelman, 1994).

**Divergence free vector fields.** First we focus on the vector fields satisfying (5) for arbitrary large dimensions. Property (6) can then be incorporated with very little additional work.

**Proposition 2.** *Consider an arbitrary vector field $\mathbf{v} : \mathbb{R}^d \to \mathbb{R}^d$. Then $\nabla \cdot \mathbf{v} = 0$ if and only if there exists smooth scalar functions $\psi_j^i : \mathbb{R}^d \to \mathbb{R}$, with $\psi_j^i = -\psi_i^j$ such that*

$$v_i(\mathbf{x}) = \sum_{j=1}^{d} \partial_{x_j} \psi_j^i(\mathbf{x}), \quad i = 1, ..., d, \tag{7}$$

*where $\mathbf{v} = (v_1, ..., v_d)$.*

To impose the boundary conditions (6) one can simply multiply each $\psi_j^i$ by $(x_i^2 - 1)(x_j^2 - 1)$.

**Lemma 2.** *Consider the coefficients*

$$\psi_j^i(\mathbf{x}) = h_i(x_i) h_j(x_j) \widetilde{\psi_j^i}(\mathbf{x}) \tag{8}$$

*where $h_i(x_i) = h_i^1(x_i - 1)h_i^2(x_i + 1)$, $h_i^1$, $h_i^2$ are functions satisfying $h_i^1(0) = h_i^2(0) = 0$ and $\widetilde{\psi_j^i}(\mathbf{x}) : \mathbb{R}^d \to \mathbb{R}$ are bounded functions satisfying $\widetilde{\psi_j^i} = -\widetilde{\psi_i^j}$. Then the function $\mathbf{v}$ defined in Proposition 2 satisfies $\nabla \cdot \mathbf{v} = 0$ and $\mathbf{v} \cdot \mathbf{n} = 0$ on $\partial\Omega$.*

The function $\mathbf{h} = (h_1, ..., h_d)$ can typically be parametrized as neural networks with no bias but in our experiments we have simply chosen to consider the case $h_i(x_i) = (x_i^2 - 1)$. One drawback of Lemma 2 is that it does not guarantee a universal approximation of divergence free functions near the boundaries. Note that when we are away from the boundaries however Proposition 2 gives this universal approximation result. In our experiments we observed that we can at least significantly reduce the OT cost with the construction (8) and we therefore let the investigations related to the boundary conditions for future work.

The incompressible property (5) and the boundary conditions (6) can be exactly implemented in the network in any dimension. Note however that in order to get all the incompressible vector fields (7), we need to construct at least $d(d-1)/2$ arbitrary scalar functions. See Appendix A for the practical construction of these divergence free functions in high dimensions.

## 4 Euler's geodesics

GP flows give a way to compute the Monge map for any trained NF architecture. Many transformations can achieve this goal and the question of finding the best flow among all volume preserving transformations needs to be considered. In the following we will consider a regularization term to smooth the trajectories by minimizing the energy $\int \mathbf{v}^2$. While it could be tempting to try to penalize directly $\int \mathbf{v}^2$, the energy will unfortunately have an opposite objective from the minimization of the OT cost. Indeed the global minimum for the energy is $\mathbf{v}^2 = 0$ (that is the particles do not move) which is obviously not the velocity field which minimizes the OT cost. In fact the global minimum of a loss composed with the two terms OT $+ \int \mathbf{v}^2$ may not minimizes the OT cost. On the contrary, as we will detail below, Euler's equations minimize the energy on SDiff over all other solutions with the same initial and final states. By finding the correct final state and solving Euler equations it is therefore possible to obtain a solution which both minimizes the OT cost and the energy.

### 4.1 Arnold's theorem

In 1966, Arnold (Arnold, 1966) showed that the flow described by Euler's equations coincides with the geodesic flow on the manifold of volume preserving diffeomorphisms. This theoretical result therefore gives the reason why regularizing our flows with Euler's equations is a desirable property. Mainly that Euler's equations take the path with the lowest energy to reach the final configuration. Consider the Euler equations:

$$\begin{cases} \partial_t \mathbf{v} + (\mathbf{v} \cdot \nabla)\mathbf{v} = -\nabla p, & t \in [0, T], \ \mathbf{x} \in \Omega, \\ \nabla \cdot \mathbf{v} = 0, & t \in [0, T], \ \mathbf{x} \in \Omega, \\ \mathbf{v} \cdot \mathbf{n} = 0, & t \in [0, T], \ \mathbf{x} \in \partial\Omega, \\ \mathbf{v}(0, \cdot) = \mathbf{v}_0, \end{cases} \tag{9}$$

where $\mathbf{v} := \mathbf{v}(t, \mathbf{x})$ is the velocity field, $p := p(t, \mathbf{x})$ the pressure and $\mathbf{n} := \mathbf{n}(\mathbf{x})$ the outward normal at the boundary of $\Omega$. Here the pressure $p$ ensures that $\partial_t \mathbf{v} + (\mathbf{v} \cdot \nabla)\mathbf{v}$ can be written as the gradient of some scalar function which is uniquely defined (up to a constant) thanks to the additional divergence free and boundary conditions on $\mathbf{v}$. Additionally the pressure field can be interpreted as the Lagrange multiplier of the divergence free constraint for the associated variational formulation of Euler's equations. In particular, it may not be needed to compute $p$ when solving numerically (9). We introduce $\mathcal{E}$ the energy of a smooth function $\mathbf{X}(t, \cdot)$:

$$\mathcal{E}(\mathbf{X}) = \int_0^T \int_\Omega \frac{1}{2} |\partial_t \mathbf{X}(t, \mathbf{x})|^2 d\mathbf{x} dt, \tag{10}$$

Now assume $\phi \in \mathrm{SDiff}(\Omega)$. Arnold's problem's consists in finding the path $\mathbf{X}(t, \cdot)_{t \in [0, T]}$ in $\mathrm{SDiff}(\Omega)$ joining the identity to $\phi$ which minimizes $\mathcal{E}$:

$$\min_{\mathbf{X}(t, \cdot) \in \mathrm{SDiff}(\Omega)} \mathcal{E}(\mathbf{X}), \quad \mathbf{X}(0, \cdot) = \mathrm{Id}, \quad \mathbf{X}(T, \cdot) = \phi(\cdot). \tag{11}$$

In other words (11) is the geodesic in $\mathrm{SDiff}(\Omega)$ between Id and $\phi$.

**Theorem 2** (Arnold (1966)). *Assuming the existence of a solution to Arnold's problem, $\mathbf{X}$ is solution to* (11) *if and only if $\mathbf{v}(t, \mathbf{x}) := \partial_t \mathbf{X}(t, \mathbf{x})$ satisfies Euler's equations* (9).

### 4.2 Penalization of Euler's equations in high dimensions

Numerical schemes developed to efficiently solve the Euler equations (Canuto et al., 2007; Quarteroni, 2009) (mainly for fluid mechanics problems, i.e. for dimensions up to 3) scale badly when the dimension increases. In this work, the solution to Euler's equations is interpreted as the geodesic to reach the solution of the OT problem and the dimension can be arbitrary large. Therefore we approach the equation (9) through a penalization procedure which can be carried out in any dimension. As explained in the previous section we notice that the second and third equations in (9) are satisfied by construction in the network.

Our remaining goal is to constrain the network to be a smooth solution to $\partial_t \mathbf{v} + (\mathbf{v} \cdot \nabla)\mathbf{v} = -\nabla p$. The left hand side can therefore be written as the gradient of a scalar function and we note that a vector satisfies $\mathbf{w}_{t,\mathbf{x}} \in \mathbb{R}^d$ satisfies $\mathbf{w}_{t,\mathbf{x}} = \nabla p(t,\mathbf{x})$ if and only if its Jacobian is symmetric $\nabla \mathbf{w}_{t,\mathbf{x}} = (\nabla \mathbf{w}_{t,\mathbf{x}})^T$. In order to solve the first equation in (9), we propose to penalize the non-symmetric part of the Jacobian for the total derivative of $\mathbf{v}$. Since a Jacobian-vector product can be efficiently evaluated in high dimensions (unlike the calculation of the full Jacobian which is computationally expensive), we do not calculate directly the Jacobian and use instead the following property of symmetric matrices: $M$ is symmetric if and only if $\mathbf{y}^T M \mathbf{z} - \mathbf{z}^T M \mathbf{y} = 0$, $\forall \mathbf{y}, \mathbf{z} \in \mathbb{R}^d$. The idea is to sample random vectors $\mathbf{y}, \mathbf{z}$ during the training and to penalize this term for the total derivative, that is to minimize:

$$R(\mathbf{x}) := \mathbb{E}_{\mathbf{y},\mathbf{z}} \left[ \int_0^T \left( \mathbf{y}^T (\nabla \mathbf{w}_{t,\mathbf{x}}) \mathbf{z} - \mathbf{z}^T (\nabla \mathbf{w}_{t,\mathbf{x}}) \mathbf{y} \right)^2 dt \right], \tag{12}$$

with $\mathbf{y}, \mathbf{z} \sim \mathcal{N}(\mathbf{0}, \mathrm{Id})$ and $\mathbf{w}_{t,\mathbf{x}} = \partial_t \mathbf{v} + (\mathbf{v} \cdot \nabla)\mathbf{v}$. In practice, we do not compute the full time integral in (12) as it would be computationally too expensive but calculate the penalization only at our time steps discretization.

**Approximation of the total derivative.** To reduce the computational burden, we do not calculate exactly the total derivative $\mathbf{w}_{t,\mathbf{x}}$ but use an approximation of its Lagrangian formulation instead. More precisely, consider the variable $\mathbf{X}(t,\mathbf{x})$ from (4) that is the position of a particle at time $t$ with initial position $\mathbf{x}$. We recall the equality (see Appendix B.3.1) $\frac{D}{Dt}\mathbf{v}(t,\mathbf{X}(t,\mathbf{x})) = \partial_t \mathbf{v}(t,\mathbf{X}(t,\mathbf{x})) + (\mathbf{v}(t,\mathbf{X}(t,\mathbf{x})) \cdot \nabla)\mathbf{v}(t,\mathbf{X}(t,\mathbf{x}))$ and therefore choose to approximate the right-hand side by using a first order Taylor expansion of $D\mathbf{v}/Dt$:

$$\frac{D}{Dt}\mathbf{v}(t,\mathbf{X}(t,\mathbf{x})) \approx \frac{\mathbf{v}(t^{n+1}, \mathbf{X}(t^{n+1}, \mathbf{x})) - \mathbf{v}(t^n, \mathbf{X}(t^n, \mathbf{x}))}{\Delta t}, \tag{13}$$

$\Delta t := t^{n+1} - t^n$. In practice, $\Delta t$ is set to $2\sqrt{\varepsilon}$ where $\varepsilon$ is the machine precision. This approximation can be easily computed since it requires only the evaluation of the velocity at two positions of a particle.

To summarize our approach requires to penalize a Jacobian-vector product of the form $\mathbf{y}^T(\nabla \mathbf{w})\mathbf{z} - \mathbf{z}^T(\nabla \mathbf{w})\mathbf{y}$ where $\mathbf{w}$ is given by (13). Penalizing Jacobian-vector product has already been done in other contexts and prove to efficiently scale with the dimension Song et al. (2020).

## 5 Procedure

In order to solve the optimal transport problem, we can either use the forward NF function $\mathbf{f}$ or $\mathbf{g} := \mathbf{f}^{-1}$. Depending on this choice the loss function is then either $E_{\mu(\mathbf{x})}\|\mathbf{x} - \mathbf{s} \circ \mathbf{f}(\mathbf{x})\|^2$ or $E_{\nu(\mathbf{x})}\|\mathbf{x} - \mathbf{g} \circ \mathbf{s}(\mathbf{x})\|^2$.

In practice the GP flow is parametrized as a standard residual network (ResNet) with a Runge-Kutta 4 time discretization (Atkinson, 1989) (other discretization are possible) and is estimated by minimizing the parameters of the velocity field. When regularizing with Euler's equations, we replace the term $\nabla \mathbf{w}$ in (12) by (13) and calculate the Jacobian-vector product with the function *torch.autograd* from pytorch. A parameter $\lambda > 0$ is also added in front of the penalization term that is if we use the function $\mathbf{f}$:

$$\min_{\boldsymbol{\theta}} E_{\mu(\mathbf{x})} \left[ \|\mathbf{x} - \mathbf{s}_{\boldsymbol{\theta}} \circ \mathbf{f}(\mathbf{x})\|^2 + \lambda R_{\boldsymbol{\theta}} \circ \mathbf{erf} \circ \frac{\mathbf{f}(\mathbf{x})}{\sqrt{2}} \right], \tag{14}$$

or if the function $\mathbf{g}$ is considered instead

$$\min_{\boldsymbol{\theta}} E_{\nu(\mathbf{x})} \left[ \|\mathbf{x} - \mathbf{g} \circ \mathbf{s}_{\boldsymbol{\theta}}(\mathbf{x})\|^2 + \lambda R_{\boldsymbol{\theta}} \circ \mathbf{erf} \circ \frac{\mathbf{x}}{\sqrt{2}} \right], \tag{15}$$

where the vector $\boldsymbol{\theta}$ denotes the parameters of the velocity field $\mathbf{v}$, $\mathbf{f}$ is the NF architecture, $\mathbf{s}$ the GP flow and $R$ corresponds to the term penalized with Euler's equations. The subscript $\boldsymbol{\theta}$ has been added to highlight the dependence of $\mathbf{s}$ and $R$ to the parameters. We emphasize that (14)-(15) are two distinct optimization strategies:

- If we do not want to invert the NF model (for example if it is computationally expensive) we can simply minimize (14) over the training data. The probability distribution $\mu$ in (14) is then the unknown distribution from which the data are taken. This assumes however that the training data are correctly mapped to the standard distribution $\nu$ with $\mathbf{f}$.

- If the NF model is cheap to invert we can minimize (15) instead. To do so we optimize over samples from the standard normal distribution $\nu$. In this case, the probability distribution $\mu$ is the transformation of $\nu$ by the inverse of the NF model $\mu = \mathbf{g}_\# \nu$.

If the NF model transforms perfectly the training data over the standard normal $\nu$, these two approaches are equivalent. If this is not the case, we notice that the second approach requires a cheap inverse of the NF model, but has the advantage of not using any training data to train the GP flow and may therefore better generalize.

## 6 Results

We apply GP flows on two popular NF models: BNAF, a discrete NF (De Cao et al., 2020) for two-dimensional test cases and FFJORD, a continuous NF (Grathwohl et al., 2018) for higher dimensional cases. Both of these models are solid references among NF and do not incorporate any OT knowledge in their architecture or training procedure. The codes are taken from the official repositories[1]. The FFJORD model has an inverse function directly available in the code, which is not the case for the BNAF model. For this reason we consider only the FFJORD model when interpolating in the latent space of the dSprites and MNIST datasets because interpolations require the NF architecture to have an inverse function available. To compare our results we consider the CP-Flow architecture (Huang et al., 2020). The CP-flow network is constrained to be the composition multiple blocks which are gradient of scalar convex function and therefore converges by construction towards the optimal map when considering only 1 block (provided the optimization reaches a global minimum). This makes CPFlow a good candidate for comparison.

### 6.1 Density estimation on toy 2D data

In this section we perform density estimation on several 2d standard toy distributions (Grathwohl et al., 2018; Wehenkel & Louppe, 2019). In particular we train our GP flow on a pre-trained BNAF model. We provide two dimensional toy examples for the eight Gaussians, two moons and pinwheel test case on Figure 3. In the case of the pinwheel dataset we use Euler regularization see below for more details. To compute the exact discrete OT projection we use the POT library Flamary et al. (2021). The distribution is colored to compare the transformation with the exact OT map. We observe on each experiment that adding GP flow makes the transformation closer to the Monge map.

**Euler regularization.** Let's briefly recall the goal when considering Euler's regularization. As discussed in Theorem 1 the measure preserving transformation $\mathbf{s}$ which minimizes the OT cost is unique. In this work we have constructed $\mathbf{s}$ to be the solution of an ODE with divergence free velocity and therefore even if $\mathbf{s}$ (that is the solution of the ODE at the final time) is unique, there are infinitely many trajectories which reach this final configuration. The role of Euler regularization is to obtain smooth trajectories for our ODE, a property which may be helpful during the training process to minimize the OT cost.

To highlight the value of using the Euler's penalization we focus on the pinwheel test case. If the pinwheel experiment from Figure 3 was done with Euler regularization, this is because we notice a bad convergence of the GP flow without it. Figure 4 shows the reason why the training has trouble to converge: the trajectories are very irregular when no penalization is applied. This is an example of the practical utilization of Euler's

---

[1] github.com/rtqichen/ffjord, github.com/nicola-decao/BNAF, github.com/CW-Huang/CP-Flow

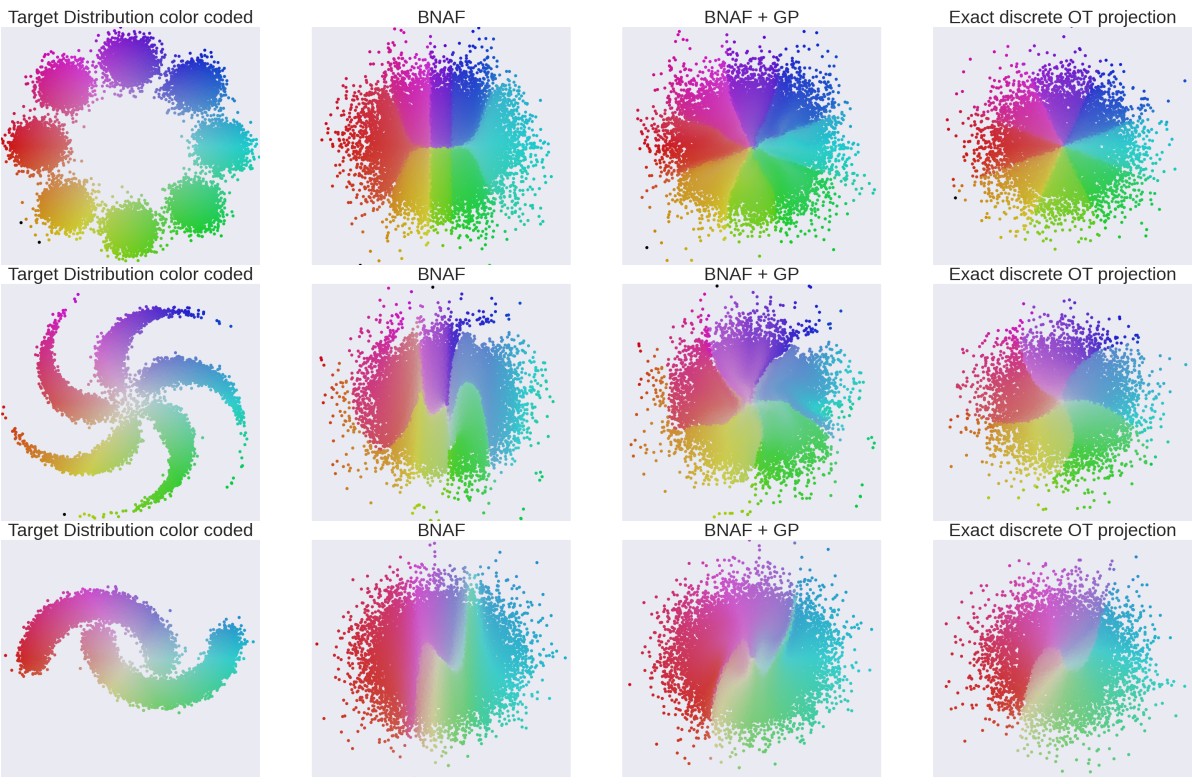

Figure 3: Color map for several 2d toy examples. Adding GP flow makes the transformation closer to the OT map.

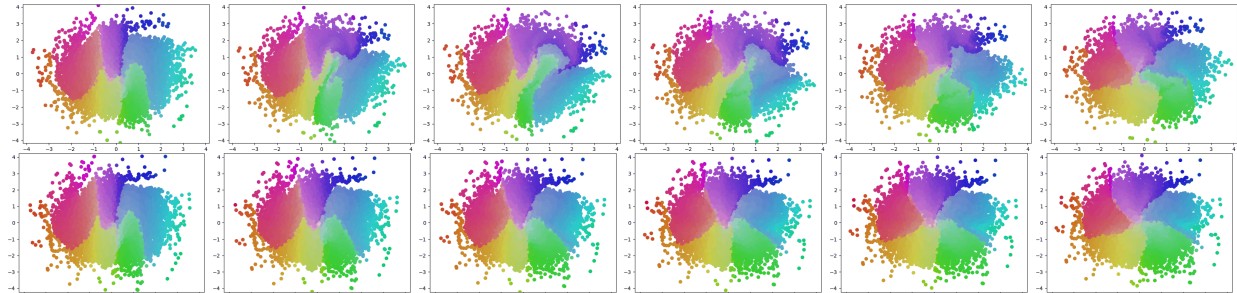

Figure 4: Trajectories for the pinwheel test case from $T = 0$ (leftmost image) to $T = 1$ (rightmost image). Top: Gaussian motion without regularization. Bottom: Gaussian motion with Euler regularization. The latter gives much smoother trajectories.

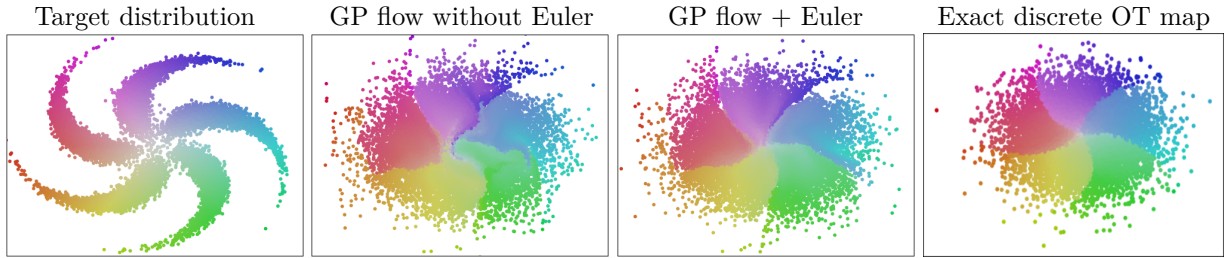

Figure 5: Comparison of GP flow with and without Euler for the Pinwheel test case. Euler regularization leads to a better convergence result.

No regularization        Euler regularization

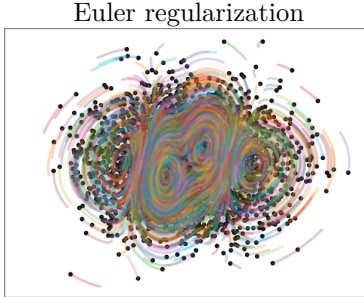

Figure 6: Representation of the trajectories of the GP flow for the two moons test case with and without Euler penalization. Trajectories obtained with Euler regularization are more structured.

regularization and as shown in Figure 5 it leads to a better global convergence. We give another illustration of the trajectories for the two moons test case on Figure 6. We observe that the trajectories obtained with Euler regularization are more structured.

Note that Euler regularization is not always needed to reach the OT map and in practice it seems to depend on the case considered. More details on Euler penalization are given in appendix D.

## 6.2 An example of data structure preservation: improving disentanglement preservation with optimal transport

The goal here is to show that OT transformations better preserve the data structure than non OT ones. Preserving data structure can be very useful for example when trying to interpolate between data points. Indeed, interpolating directly in the data space often leads to poor results mainly because we may interpolate through low density regions. In this case it can be a good idea to map the data density into a Gaussian, interpolate in the Gaussian space and then apply the reverse transformation. However, for the interpolation to be meaningful we expect the NF architecture to preserve as much of the data structure as possible. In order to study high dimensional data structure we therefore seek a data space which satisfy two properties:

- The initial data space should be very structured.

- The interpolations between data points should be easily comparable and in particular we should be able to classify them easily.

Disentangled representations satisfy these two properties and allow to encode the data in a latent space where change in one direction result in the change over one generative factor in the data. Recently the construction of variational auto-encoders (VAE) (Kingma & Welling, 2014) with disentangled latent space has received much attention (Higgins et al., 2016; Burgess et al., 2018; Chen et al., 2018a; Kim & Mnih, 2018). We therefore propose to experimentally study disentanglement preservation of NF with and without OT. To do this we apply a NF architecture (FFJORD in our case) to the VAE's latent distribution and consider the addition of a GP flow. The goal here is therefore to get as close as possible to the initial interpolations in order to show that even if the initial density is transformed to a Gaussian the data structure is preserved at least to some extent.

For the disentangled interpolation in the NF target (Gaussian) space we consider the same directions which are present in the VAE latent space and are aligned with the axes. This may be a little naive since these directions could be slightly modified depending on the source and target distributions, but it seems a good enough approximation here. Our experiments are run with the $\beta$-TCVAE architecture (Chen et al., 2018a) and the latent space dimension is 10.

**dSprites dataset.** The dSprites dataset (Matthey et al., 2017) is made of $64 \times 64$ images of 2D shapes procedurally generated from 5 ground truth independent latent factors. These factors are shape, scale, rotation, x and y positions of a sprite. Since the factors are known, we can compute a quantitative evaluation of disentanglement and we choose here to consider the metric from (Eastwood & Williams, 2018) on the continuous factors (i.e. all the factors except the shape) for the three criteria: disentanglement,

| Model | Disentenganlement ↑ | Completeness ↑ | Informativness ↓ |
|---|---|---|---|
| Init. latent space | **0.58** | **0.81** | **0.55** |
| FFJORD | 0.39 | 0.26 | 0.62 |
| FFJORD+GP | **0.58** ± 0.01 | 0.66 ± 0.01 | 0.62 ± 0. |
| FFJORD+GP+EULER | **0.58** ± 0.01 | 0.66 ± 0.01 | 0.58 ± 0. |

Table 1: Quantitative evaluation from (Eastwood & Williams, 2018) of disentanglement (higher is better), completeness (higher is better) and informativeness (lower is better) on the dSprites dataset. Adding GP flows make the scores closer to the initial ones. For all models considered here (except in the case of the vanilla FFJORD model) the values represent the mean and standard deviation taken over 3 runs.

| Model | dSprites | MNIST | chairs |
|---|---|---|---|
| **OT cost** | | | |
| FFJORD | 10.45 | 6.81 | 5.98 |
| FFJORD+GP | 5.69 ± 0.03 | **3.11** ± 0.01 | 2.34 ± 0.01 |
| FFJORD+GP+EULER | **5.30** ± 0.01 | **3.11** ± 0.01 | 2.36 ± 0.01 |
| CPFlow (3 blocks) | 6.27 ± 0.06 | 6.01 ± 2.12 | 2.46 ± 0.14 |
| CPFlow (1 block) | 8.97 ± 3.44 | 27.92 ± 3.00 | **2.21** ± 0.50 |
| **Loss** | | | |
| FFJORD | −16.62 | **-0.45** | **5.48** |
| FFJORD+GP | −16.62 ± 0. | **-0.45** ± 0. | **5.48** ± 0. |
| FFJORD+GP+EULER | −16.62 ± 0. | **-0.45** ± 0. | **5.48** ± 0. |
| CPFlow (3 blocks) | **-19.07** ± 0.06 | −0.06 ± 0.05 | 5.94 ± 0.03 |
| CPFlow (1 block) | −16.99 ± 0.08 | 0.95 ± 0.11 | 6.45 ± 0.01 |

Table 2: Losses and mean OT costs. GP flows reduce the OT cost without changing the loss. Adding Euler regularization allows to further reduced the OT cost on the dSprites dataset. For all models considered here (except in the case of the vanilla FFJORD model) the values represent the mean and standard deviation taken over 3 runs.

completeness and informativeness. Table 1 shows that FFJORD destroys the latent structure and gives the worst disentanglement, completeness and informativeness scores. Adding GP flows allow to recover the same disentanglement score as the initial latent space and get values closer both for completeness and informativeness. On Table 2 the OT costs are compared, and as expected GP flows allow to reduce the OT cost without changing the loss. Interestingly GP flows with no additional regularization do not converge completely to the Monge map because particles get out of the domain at some point, making it impossible to continue the training process (the training is stopped at ≈ 60 epochs). We conjecture that this may be due to non-smooth trajectories of our GP flows and a regularization is therefore needed. As shown on Table 2 adding Euler regularization fixed this issue and allows to further reduce the OT cost.

To illustrate the preservation of disentanglement, some interpolations are also presented in Figures 9 and 10 in Appendix E. The dimensions are sorted with respect to their KL divergence in the initial latent space and therefore only the first dimensions carry information: each of the first 5 lines correspond to a generative factor while the last dimensions leave the image unchanged. This structure is lost when mapping the latent space to a Gaussian distribution with the FFJORD architecture. The addition of a GP flow fixed this issue and the interpolation better match the initial latent one.

**MNIST dataset.** We also consider the MNIST dataset (Lecun et al., 1998). As opposed to the dSprites test case, GP flows do not seem to have trouble to converge here without Euler penalization and therefore we obtain comparable OT costs with and without Euler regularization see Table 2. Since the generative factors are not known in this case we cannot make a quantitative evaluation of disentanglement as we did for the dSprites dataset. We focus instead on the interpolations presented in Figures 11 and 12 in Appendix E. GP

flows better preserve the data structure of the initial latent space compare when applying only the FFJORD model. This can be seen in particular on the last two rows of each block which are not changing in the initial latent space. This structure is lost with the FFJORD model and recovered when training a GP flow.

**Chairs dataset.** Finally we look at the chairs dataset (Mathieu et al., 2014). Again, the GP method converges both with and without Euler's penalization and greatly reduced the OT cost see Table 2. Interpolations are presented in Figures 13 and 14 in Appendix E. By looking at the last rows we observe once again that GP flows better preserve the data structure of the initial latent space compare when applying only the FFJORD model.

**Comparison with CPFlow.** To check that GP flows get close to the Monge map we also compare the OT costs with the CPFlow architecture. Note that the number of parameters as well as the training procedure differ from the FFJORD+GP and CPFlow architectures so we only use CPFlow OT costs as a baseline to ensure that FFJORD+GP is close to the Monge map. The losses are only given here as an indication to show that the probability distribution of both FFJORD and CP flow are close from each other and a comparison between their OT costs is therefore relevant. Finally note that we both test the CP flow architecture with 1 and 3 blocks. The advantage with the 1 block architecture is that CP flow should theoretically converges to the Monge map. However it seems harder to make the network converges in this case and we therefore also show some comparisons with 3 blocks which leads to much lower losses.

GP flows get OT values which are always lower than the CP flow ones for similar losses and we therefore conclude that GP flows are at least as close to the Monge map as CPFlow. The only exception is the chairs dataset where CP flow with one block has a lower OT cost. We note however that in this case the loss for CP flow is much bigger and therefore the comparison of OT costs with GP flow may not be relevant. We also notice that CPFlow may leads to high OT cost when trained on the MNIST dataset. One possible explanation for this is that CPFlow is trained only on the data. Since the OT costs are evaluated with random samples drawn from the standard normal this may show some issue with the generalization during the backward process. On the contrary GP flows are directly trained on random samples from this distribution and may therefore better generalize.

It is therefore possible to better preserve the data structure with GP flows by significantly reducing the OT cost. Note however that since we were not able to make the CP flow architecture converge with only 1 block in our experiments, it could be a good idea to try to assess more precisely how close GP flow is to the Monge Map. To this end there exists many other approaches which could be considered. One could for example use dedicated OT benchmarks (Korotin et al., 2021) or try more robust approaches (Korotin et al., 2022; Makkuva et al., 2020).

## 7  Discussion

This article describes a method to reduce the OT cost of any pre-trained NF model without changing the estimated target density. The proposed method has been tested up to $d = 10$ and does not require to constrain the architecture of the original model. The procedure relies on building Gaussian preserving flows to rearrange the source distribution to satisfy the OT property. The proposed approach is based on incompressible vector fields which allow to use a nice interpretation of Euler's equations as a geodesic in the group of volume-preserving diffeomorphisms between the identity and the transformation minimizing the OT cost. This original contribution allows to add a regularity condition to the estimated map in addition to simply enforcing the OT property.

**Perspectives.** The numerical experiments presented here pave the way to new research perspectives. First compared to other OT approaches in the NF literature, GP flows is to the best of our knowledge the first one which does not constrain the NF architecture to obtain the Monge map. This could be a great advantage when the NF architecture is already constrained for other reasons (for example to satisfy some symmetries or data-related properties). We believe that in this case GP flows may stand out as it could be difficult to further constrain the network to satisfy the OT property with standard approaches. The proposed approach could also be a starting point to investigate other type of (potentially non OT) costs, and more specifically non-quadratic OT costs, such as the $L_1$ norm which is much less considered in the literature due to the lack

of theoretical foundations (the OT map is not the gradient of a convex function anymore). In a GP-based framework, such extension could be easily implemented since we do not rely explicitly on this property. Finally let's mention the case where we consider a map $h$ between two unkown distribution $D_1$ and $D_2$ but none of them is Gaussian. To recover the Monge map associated with $h$ a possibility might be to map $D_1$ to a Gaussian, makes the rearrangement in the Gaussian space with respect to the quadratic cost of $h$ (that is between $D_1$ and $D_2$) and then map the points back to $D_1$. In this case we could recover the OT map between $D_1$ and $D_2$ even if none of them is Gaussian.

**Limitations.** The main limitation of the proposed method is probably related to the number of independent functions require to construct incompressible vector fields in high dimensions. Indeed, as explained in Proposition 2 one needs to construct at least $d(d-1)/2$ scalar functions to get all the divergence free vector fields in dimension $d$ and in practice our approach requires $d-1$ vector valued functions in $\mathbb{R}^d$. It could therefore be desirable to scale up more efficiently with the dimension. One possible way to overcome this difficulty would be to give up on the exact implementation of divergence free functions. For example one could add a penalization term of the divergence in the loss with an unbiased estimator of the divergence (Song et al., 2020). Also, let us mention that while Euler's regularization adds nice properties to the flows considered, it also increases the computational time required to train the model. Finally the total duration of the training can be impacted by the initial NF chosen since its evaluation is required to compute the OT cost. NF architectures with fast forward (or backward) pass should therefore be preferred.

### Acknowledgments

The research leading to these results has received funding from IMT Atlantique, Cominlabs Labex (DY-NALEARN project) and ANR (AI4CHILD, ANR-19-CHIA-0015-01, project LEMONADE, ANR-21-CE48-0005 and project OTTOPIA ANR-20-CHIA-0030).

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

## A    Practical construction of incompressible vector fields in high dimensions

The goal here is to have a GPU-friendly construction of the incompressible vector fields given in Proposition 2. In the following we consider stationary divergence free functions but the time variable can be added with no additional work simply by considering functions in $\mathbb{R}^{d+1}$ instead of $\mathbb{R}^d$ (the gradients are still taken only on the space variables though). All the proofs are given in Appendix B.

**Notations.** Regarding the notations we will use the operator diag for two distinct cases: 1) When $\mathbf{w}$ is a vector diag($\mathbf{w}$) denotes the diagonal matrix obtained from the vector $\mathbf{w}$. 2) When $W$ is a matrix diag($W$) denotes the vector obtained from the diagonal of $W$. The operation $\cdot$ denotes the scalar product between two vectors. Finally we have adopted the convention that when a scalar multiplies a vector it multiplies each of its component.

**Practical construction.** Let $\mathbf{u}^n : \mathbb{R}^d \to \mathbb{R}^{d-n}$. We construct a divergence free function with the functions $\psi_j^i$ defined as

$$(\psi_j^i)^n = \begin{cases} u_{i-n}^n - u_{j-n}^n, & \text{if } i, j \geq n+1, \\ 0, & \text{otherwise.} \end{cases} \tag{16}$$

To construct this divergence free function we define the matrix $(\nabla \mathbf{u}(\mathbf{x}))^n \in \mathbb{R}^{d \times d}$ and the vector $\mathbf{1}^n \in \mathbb{R}^d$ as

$$(\nabla \mathbf{u}(\mathbf{x}))_{ij}^n = \begin{cases} \partial_{i-n} u_{j-n}^n, & \text{if } i, j \geq n+1 \\ 0, & \text{otherwise.} \end{cases}, \tag{17}$$

$$\mathbf{1}_i^n = \begin{cases} 1, & \text{if } i \geq n+1 \\ 0, & \text{otherwise.} \end{cases}.$$

**Lemma 3.** *Let $n \in \mathbb{N}$, $n \leq d - 2$ and consider the function $\mathbf{u}^n : \mathbb{R}^d \to \mathbb{R}^{d-n}$. Then the vector field $\mathbf{v}^n : \mathbb{R}^d \to \mathbb{R}^d$ defined as*

$$\mathbf{v}^n(\mathbf{x}) = (\nabla \mathbf{u})^n \ \mathbf{1}^n - [\text{diag}(\nabla \mathbf{u})^n \cdot \mathbf{1}^n]\mathbf{1}^n, \tag{18}$$

*is divergence free.*

To construct the functions (18) we need 1) to compute the product between the Jacobian of a vector valued function and a constant vector 2) sum the diagonal elements of the Jacobian matrix. Both of these operations

can be done efficiently on GPU. Note that in order to satisfy the boundary conditions $\mathbf{v}^n \cdot \mathbf{n} = 0$ one can modify the equation (18) as in Lemma 2 (recalling we are taking in practice $h_i = x_i^2 - 1$) to obtain

$$
\begin{aligned}
\mathbf{v}^n(\mathbf{x}) = (\mathbf{x}^2 - 1)\odot \\
\left[ 2M^n\mathbf{x} + (\nabla\mathbf{u})^n(\mathbf{x})(\mathbf{x}^2 - 1) - ((\mathbf{x}^2 - 1) \cdot \mathrm{diag}(\nabla\mathbf{u}(\mathbf{x}))^n)\mathbf{1}^n \right],
\end{aligned}
\tag{19}
$$

where $M_{ij}^n = u_i^n - u_j^n$, if $i, j \geq n + 1$ and $M_{ij}^n = 0$ otherwise.

It is possible to recover all the incompressible functions from Proposition 2 by adding the blocks $\mathbf{v}^0 + \mathbf{v}^1 + ... + \mathbf{v}^{d-2}$.

**Proposition 3.** *Let $\mathbf{v}$ be a divergence free function in $\mathbb{R}^d$. Then there exists $d - 1$ functions $\mathbf{v}^0, ..., \mathbf{v}^{d-2}$ constructed as in Lemma 3 such that*

$$
\mathbf{v}(\mathbf{x}) = \sum_{n=0}^{d-2} \mathbf{v}^n(\mathbf{x}).
$$

The attentive reader would have noticed that with the vector functions $\mathbf{u}^n$, $n = 0, ..., d - 2$ we have a total of $(d+2)(d-1)/2$ independent scalar functions while Proposition 2 only requires the construction of $d(d-1)/2$ scalar functions leaving $d - 1$ additional functions which are not strictly needed to obtain the divergence free vectors. This is due to the vectorized constructions (18)-(19) which allow a fast evaluation of the divergence free functions on GPU. Having $d - 1$ additional scalar functions in return is not a big issue since the general order remains $O(d^2)$.

Also note that the practical implementation of the equations (16)-(17) requires to find a pythonic way to efficiently pad a group of matrices with different dimensions. We have not yet find such way and therefore have simply chosen in our applications to construct $d - 1$ vector valued functions $\mathbf{u}^n \in \mathbb{R}^d$, $n = 0, ..., d - 2$ and fill the appropriate dimensions in (16)-(17) with 0. Again even if not optimal this is not a big issue as it multiplies the number of independent scalar functions by a factor 2, but the general order remains $O(d^2)$.

In practice, we have written the functions $\mathbf{u}^n$ as the output of a big function $\mathbf{u} : \mathbb{R}^d \to \mathbb{R}^{(d-1)\times d}$ allowing to evaluate all the functions $\mathbf{u}^n$ in a single pass. The vector $\mathbf{u}$ is written as the composition of linear functions with some simple non-linearity

$$
\begin{aligned}
\mathbf{u}(\mathbf{x}) &= M_n\mathbf{x}_n + \mathbf{b}_n, \\
\mathbf{x}_i &= \sigma(M_{i-1}\mathbf{x}_{i-1} + \mathbf{b}_{i-1}), \quad i = 1, ..., n - 1, \\
\mathbf{x}_0 &= \mathbf{x},
\end{aligned}
\tag{20}
$$

where $M_i$ are rectangular matrices, $\mathbf{b}_i$ a vector field and typically we have taken $\sigma = \tanh$. One big advantage of the formulation (20) is that the Jacobian of $\mathbf{u}$ (and therefore of all the functions $\mathbf{u}^n$) can be computed analytically

$$
\begin{aligned}
\nabla\mathbf{u}(\mathbf{x}) &= M_n\nabla\mathbf{x}_n, \\
\nabla\mathbf{x}_i &= \mathrm{diag}(\sigma'(M_{i-1}\nabla\mathbf{x}_{i-1} + \mathbf{b}_{i-1}))M_{i-1},
\end{aligned}
\tag{21}
$$

for $i = 1, ..., n - 1$. The formulation (21) therefore allows a fast evaluation of the term (18) in particular when summing the diagonal elements of the Jacobian. In our experiments we have noticed that the analytical formulation of the Jacobian (21) was faster than using *torch.autograd*.

## B   Technical material

### B.1   Gaussian preserving flows

#### B.1.1   Proof of Proposition 1

We recall that here $\mathbf{erf} : \mathbb{R}^d \to \mathbb{R}^d$ is the distribution function of a one dimensional Gaussian applied component wise.

**Proposition.** *A map* $\mathbf{s}$ *is a smooth Gaussian preserving function satisfying* (3) *if and only if there exists* $\boldsymbol{\phi} : (-1,1)^d \to (-1,1)^d$ *such that* $|\det \nabla \boldsymbol{\phi}| = 1$ *and*

$$\mathbf{s}(\mathbf{x}) = \sqrt{2}\,\mathbf{erf}^{-1} \circ \boldsymbol{\phi} \circ \mathbf{erf}(\frac{\mathbf{x}}{\sqrt{2}}), \quad \mathbf{x} \in \mathbb{R}^d. \tag{22}$$

*Proof.* We recall some basic properties about the distribution function of a one dimensional Gaussian and its inverse. One has for $x \in \mathbb{R}$

$$\mathrm{erf}(x) = \frac{2}{\sqrt{\pi}} \int_0^x e^{t^2}\,dt, \quad \frac{d}{dx}\,\mathrm{erf}(x) = \frac{2}{\sqrt{\pi}} e^{-x^2}, \tag{23}$$

$$\frac{d}{dx}\,\mathrm{erf}^{-1}(x) = \frac{\sqrt{\pi}}{2} e^{(\mathrm{erf}^{-1}(x))^2}.$$

Consider the function $\boldsymbol{\phi} : (-1,1)^d \to (-1,1)^d$ defined as

$$\boldsymbol{\phi}(\mathbf{x}) = \mathbf{erf} \circ \frac{\mathbf{s}}{\sqrt{2}} \circ \sqrt{2}\,\mathbf{erf}^{-1}(\mathbf{x}). \tag{24}$$

The goal here is to show that $|\det \nabla \boldsymbol{\phi}| = 1$ then equation (22) will follows from (24). By definition

$$|\det \nabla \boldsymbol{\phi}(\mathbf{x})| := |\det \nabla \left( \mathbf{erf} \circ \frac{\mathbf{s}}{\sqrt{2}} \circ \sqrt{2}\,\mathbf{erf}^{-1}(\mathbf{x}) \right)|.$$

Applying the equalities (23) component wise and denoting $\mathbf{x} = (x_1, ..., x_d)$ one gets

$$|\det \nabla \boldsymbol{\phi}(\mathbf{x})| = |\prod_i \frac{2}{\sqrt{\Pi}} e^{-(\frac{\mathbf{s}}{\sqrt{2}} \circ \sqrt{2}\,\mathbf{erf}^{-1}(\mathbf{x}))_i^2}$$

$$\times \frac{\det \nabla \mathbf{s}(\sqrt{2}\,\mathbf{erf}^{-1}(\mathbf{x}))}{\sqrt{2}}$$

$$\times \prod_i \sqrt{2} \frac{\sqrt{\Pi}}{2} e^{(\mathbf{erf}^{-1}(\mathbf{x}))_i^2}|,$$

since the determinant of the composition is the product of the determinants. That is

$$|\det \nabla \boldsymbol{\phi}(\mathbf{x})| = |\prod_i e^{-(\frac{\mathbf{s}}{\sqrt{2}} \circ \sqrt{2}\,\mathbf{erf}^{-1}(\mathbf{x}))_i^2} \times \det \nabla \mathbf{s}(\sqrt{2}\,\mathbf{erf}^{-1}(\mathbf{x}))$$

$$\times \prod_i e^{(\mathbf{erf}^{-1}(\mathbf{x}))_i^2}|,$$

which can be written

$$|\det \nabla \boldsymbol{\phi}(\mathbf{x})| = e^{(-\|\mathbf{s}(\sqrt{2}\,\mathbf{erf}^{-1}(\mathbf{x}))\|^2 + \|\sqrt{2}\,\mathbf{erf}^{-1}(\mathbf{x})\|^2)/2}$$

$$\times |\det \nabla \mathbf{s}(\sqrt{2}\,\mathbf{erf}^{-1}(\mathbf{x}))|.$$

Finally using $|\det \nabla \mathbf{s}(\mathbf{x})| = e^{(\|\mathbf{s}(\mathbf{x})\|^2 - \|\mathbf{x}\|^2)/2}$ one obtains $|\det \nabla \boldsymbol{\phi}(\mathbf{x})| = 1$.

Following the same kind of arguments one shows the other way around that is if $\mathbf{s}(\mathbf{x}) = \sqrt{2}\,\mathbf{erf}^{-1} \circ \boldsymbol{\phi} \circ \mathbf{erf}(\frac{\mathbf{x}}{\sqrt{2}})$ with $|\det \nabla \boldsymbol{\phi}| = 1$ then $|\det \nabla \mathbf{s}(\mathbf{x})| = e^{(\|\mathbf{s}(\mathbf{x})\|^2 - \|\mathbf{x}\|^2)/2}$ that is $\mathbf{s}$ is Gaussian preserving. $\quad\square$

### B.1.2 Orientation reversing functions

In the following Lemma we prove that very orientation reversing function satisfying $\det \nabla \boldsymbol{\psi} = -1$ can be written as the composition of a volume and orientation preserving function and the function $\boldsymbol{h}(x_1, ..., x_d) = (-x_1, x_2, x_3, ..., x_d)$.

**Lemma.** *Assume* $\boldsymbol{\psi} : \mathbb{R}^d \to \mathbb{R}^d$ *is a function satisfying* $\det \nabla \boldsymbol{\psi} = -1$ *and let* $\boldsymbol{h}(x_1, ..., x_d) = (-x_1, x_2, x_3, ..., x_d)$. *Then there exists a volume and orientation preserving function* $\boldsymbol{\phi}$ *such that* $\boldsymbol{\psi} = \boldsymbol{\phi} \circ \boldsymbol{h}$.

*Proof.* We define $\boldsymbol{\phi}$ as $\boldsymbol{\phi} = \boldsymbol{\psi} \circ \boldsymbol{h}$. This function is indeed volume and orientation preserving since it satisfies $\det \nabla \boldsymbol{\phi} = \det \nabla \boldsymbol{\psi} \det \nabla \boldsymbol{h} = 1$. By noticing $\boldsymbol{h} \circ \boldsymbol{h} = \mathrm{Id}$ one gets $\boldsymbol{\psi} = \boldsymbol{\phi} \circ \boldsymbol{h}$. $\quad\square$

### B.1.3 Proof of Lemma 1

**Lemma.** *Assume the Monge map $\mathbf{m}$ and the NF architecture $\mathbf{g}$ are $C^1$ diffeomorphisms. Then the corresponding GP flow $\mathbf{s}$ is $C^1$, the associated function $\phi$ is also $C^1$ and either satisfies $\det \nabla \phi(\mathbf{x}) = 1$ everywhere or $\det \nabla \phi(\mathbf{x}) = -1$ everywhere.*

*Proof.* The definition of $\mathbf{s} := \mathbf{m} \circ \mathbf{g}^{-1}$ ensures that $\mathbf{s}$ is indeed $C^1$. Moreover since $\mathbf{g}$ is invertible either $\det \nabla \mathbf{g} > 0$ everywhere or $\det \nabla \mathbf{g} < 0$ everywhere (if $\det \nabla \mathbf{g}(x) = 0$ this would mean that g is not invertible at this point) and the same argument applies to $\mathbf{g}^{-1}$ and to the Monge map $\mathbf{m}$. Therefore the equality $\mathbf{s} = \mathbf{m} \circ \mathbf{g}^{-1}$ implies that either $\det \nabla \mathbf{s} > 0$ everywhere or $\det \nabla \mathbf{s} < 0$ everywhere. The equality $\phi = \mathbf{erf} \circ \frac{\mathbf{s}}{\sqrt{2}} \circ \sqrt{2}\,\mathbf{erf}^{-1}$ shows that $\phi$ is also $C^1$ and that the sign of $\det \nabla \phi$ does not change. $\qquad\square$

### B.2 Divergence free functions

### B.2.1 Proof of Proposition 2

**Proposition.** *Consider an arbitrary vector field $\mathbf{v} : \mathbb{R}^d \to \mathbb{R}^d$. Then $\nabla \cdot \mathbf{v} = 0$ if and only if there exists smooth scalar functions $\psi_j^i : \mathbb{R}^d \to \mathbb{R}$, with $\psi_j^i = -\psi_i^j$ such that*

$$v_i(\mathbf{x}) = \sum_{j=1}^{d} \partial_{x_j} \psi_j^i(\mathbf{x}), \quad i = 1, ...d, \tag{25}$$

*where $\mathbf{v} = (v_1, ..., v_d)$.*

*Proof.* 1) First we prove that $\nabla \cdot \mathbf{v} = 0$. The divergence of $\mathbf{v}$ can be written

$$\nabla \cdot \mathbf{v} = \sum_i \partial_{x_i} \sum_j \partial_{x_j} \psi_j^i.$$

Using $\psi_j^i = -\psi_i^j$ one has

$$\nabla \cdot \mathbf{v} = \sum_i \left( \sum_{\substack{j \\ j<i}} \partial_{x_i} \partial_{x_j} \psi_j^i - \sum_{\substack{j \\ j>i}} \partial_{x_i} \partial_{x_j} \psi_i^j \right), \tag{26}$$

For the term $\sum_i \sum_{\substack{j \\ j<i}} \partial_{x_i} \partial_{x_j} \psi_j^i$ on the left hand side one can sum over the index $j$ first instead of the index $i$ that is

$$\sum_i \sum_{\substack{j \\ j<i}} \partial_{x_i} \partial_{x_j} \psi_j^i = \sum_j \sum_{\substack{i \\ i>j}} \partial_{x_i} \partial_{x_j} \psi_j^i.$$

Injecting this equality in (26) one gets

$$\nabla \cdot \mathbf{v} = \sum_j \sum_{\substack{i \\ i>j}} \partial_{x_i} \partial_{x_j} \psi_j^i - \sum_i \sum_{\substack{j \\ j>i}} \partial_{x_i} \partial_{x_j} \psi_i^j = 0.$$

2) Now we prove that every vector field satisfying $\nabla \cdot \mathbf{v} = 0$ can be written under the form (25). The proof from Stephen Montgomery-Smith is available online[2] for completeness we rewrite it here. The proof is made by induction with the following assumption.

**Assumption 1.** *Let $k \in \mathbb{N}$, $k \le d$. Given a smooth vector field $\mathbf{v}$ such that $\mathrm{div}_k\,\mathbf{v} := \sum_{i=1}^{k} v_i = 0$, there exists scalar functions $\psi_j^i : \mathbb{R}^d \to \mathbb{R}$, $1 \le i, j \le k$ with $\psi_j^i = -\psi_i^j$ such that $v_i = \sum_j \partial_j \psi_j^i$.*

---

[2] https://math.stackexchange.com/questions/578898

The Assumption 1 is trivial for $k = 0$. Suppose it is true for $k - 1$ we prove it for $k$: assume $\mathrm{div}_k \mathbf{v} = 0$ and let

$$f_1(x_1, ..., x_n) = \int_0^{x_1} \partial_k v_k(\xi, x_2, ..., x_n) d\xi. \tag{27}$$

Since $\partial_1 f_1 = \partial_x v_k$ one has

$$\partial_1(v_1 + f_1) + \partial_2 v_2 + ... + \partial_{k-1} v_{k-1} = 0.$$

Thanks to Assumption 1 there exists functions $\psi_j^i$ with $\psi_j^i = -\psi_i^j$ such that

$$v_1 + f_1 = \sum_{j=1}^{k-1} \partial_j \psi_j^1, \quad v_i = \sum_{j=1}^{k-1} \partial_j \psi_j^i, \text{ for } 2 \le i \le k - 1. \tag{28}$$

Now we define

$$\begin{aligned} f_2(x_1, ..., x_d) = &\int_0^{x_1} v_k(\xi, x_2, ..., x_{k-1}, 0, x_{k+1}..., x_d) d\xi \\ &- \int_0^{x_k} f_1(x_1, ..., x_{k-1}, \xi, ..., x_d) d\xi, \end{aligned} \tag{29}$$

then

$$\partial_k f_2 = -f_1, \tag{30}$$

and using (27) in (29) one gets

$$\begin{aligned} \partial_1 f_2 = &v_k(x_1, ..., x_{k-1}, 0, ..., x_d) \\ &- \int_0^{x_k} \partial_k v_k(x_1, ..., x_{k-1}, \xi, ..., x_d) d\xi = -v_k. \end{aligned} \tag{31}$$

Now we extend the functions $\psi_j^i$, $1 \le i, j \le k - 1$ by defining $\psi_k^1 = -\psi_1^k = f_2$ and $\psi_k^i = -\psi_i^k = 0$ for $2 \le i \le k$. Then extending the equations (28) with $k$ one has

$$\sum_{j=1}^k \partial_j \psi_j^1 = v_1 + f_1 + \partial_k f_2 = v_1, \quad \sum_{j=1}^k \partial_j \psi_j^i = v_i,$$

for $2 \le i \le k - 1$ and where we used the equality (30) in the first equation. Moreover with the definition of the function $\psi_j^k$ one has

$$\sum_{j=1}^k \partial_j \psi_j^k = -\partial_1 f_2 = v_k,$$

thanks to (31). This proves Assumption 1 for $k$. $\qquad\square$

### B.2.2 Proof of Lemma 2

**Lemma.** *Let $\Omega = [-1, 1]^d$ and consider the coefficients*

$$\psi_j^i(\mathbf{x}) = h_i(x_i) h_j(x_j) \widetilde{\psi_j^i}(\mathbf{x})$$

*where $h_i(x_i) = h_i^1(x_i - 1) h_i^2(x_i + 1)$, $h_i^1$, $h_i^2$ are functions satisfying $h_i^1(0) = h_i^2(0) = 0$ and $\widetilde{\psi_j^i}(\mathbf{x}) : \mathbb{R}^d \to \mathbb{R}$ are bounded functions. Then the function $\mathbf{v}$ defined in Proposition 2 satisfies $\nabla \cdot \mathbf{v} = 0$ and $\mathbf{v} \cdot \mathbf{n} = 0$ on $\partial\Omega$.*

*Proof.* Indeed since $\psi_i^i = 0$ there is no $\partial_{x_i}$ term which appear in the sum of (7) for the component $v_i$. The term $h_i(x_i)$ can therefore be factored that is $v_i = 0$ if $x_i = \pm 1$. Hence $\mathbf{v} \cdot \mathbf{n} = 0$ on $\partial\Omega$. $\qquad\square$

### B.2.3  Proof of Lemma 3

**Lemma.** *Let $n \in \mathbb{N}$, $n \leq d-2$ and consider the function $\mathbf{u}^n : \mathbb{R}^d \to \mathbb{R}^{d-n}$. Then the vector field $\mathbf{v}^n : \mathbb{R}^d \to \mathbb{R}^d$ defined as*

$$\mathbf{v}^n(\mathbf{x}) = (\nabla \mathbf{u})^n \; \mathbf{1}^n - [\mathrm{diag}(\nabla \mathbf{u})^n \cdot \mathbf{1}^n] \mathbf{1}^n, \tag{32}$$

*is divergence free.*

*Proof.* As explain in Appendix A the formulation (32) is equivalent to consider $\mathbf{v}^n$ under the form (7) with $\psi_j^i$ defined as in (16). From Proposition 2 the function $\mathbf{v}^n$ is divergence free. $\square$

### B.2.4  Proof of Proposition 3

**Proposition.** *Let $\mathbf{v}$ be a divergence free function in $\mathbb{R}^d$. Then there exists $d-1$ functions $\mathbf{v}^0, ..., \mathbf{v}^{d-2}$ constructed as in Lemma 3 such that*

$$\mathbf{v}(\mathbf{x}) = \sum_{n=0}^{d-2} \mathbf{v}^n(\mathbf{x}). \tag{33}$$

*Proof.* Let $\mathbf{v}(\mathbf{x})$ be a divergence free function and denote $(\psi_j^i)$ its coefficients from (7). We construct the associate vectors $\mathbf{u}^n$ of the functions $\mathbf{v}^n$ procedurally: for $k = 1, ..., d-1$, we iteratively chose $u_1^{k-1}$ arbitrarily and define the other components of the vector $\mathbf{u}^{k-1}$ as

$$u_{j-k+1}^{k-1} = u_1^{k-1} - \sum_{n=0}^{k-2}(u_{k-n}^n - u_{j-n}^n) + \psi_j^k, \quad j \geq k+1, \tag{34}$$

note that the sum is well-defined because the components $u^k$ are constructed iteratively from $k = 1$ to $k = d-1$ and we have adopted the convention $\sum_{n=0}^{-1} = 0$. We claim that with this construction we recover the equality (33). Indeed from (34) one has

$$\psi_j^k = u_{j-k+1}^{k-1} - u_1^{k-1} + \sum_{n=0}^{k-2}(u_{k-n}^n - u_{j-n}^n), \quad j \geq k+1.$$

Using (16) one gets

$$\psi_j^k = (\psi_j^k)^{k-1} + \sum_{n=0}^{k-2}(\psi_j^k)^n, \quad j \geq k+1.$$

Again using (16) one has $(\psi_j^k)^n = 0$ for $n \geq k$ and therefore

$$\psi_j^k = \sum_{n=0}^{d-2}(\psi_j^k)^n, \quad j \geq k+1.$$

Since the coefficients $\psi_j^k$ and $(\psi_j^k)^n$ are all antisymmetric this equality is also satisfied for $j < k+1$. We conclude with the decomposition (7) of the divergence free functions. $\square$

### B.3  Additional material

### B.3.1  Total derivative expansion

In this subsection we recall the expansion of the Lagrangian derivative $\frac{D}{Dt}\mathbf{v}(t, \mathbf{X}(t, \mathbf{x})) = \frac{d}{dt}\mathbf{v}(t, \mathbf{X}(t, \mathbf{x})) + (\mathbf{v}(t, \mathbf{x}) \cdot \nabla)\mathbf{v}(t, \mathbf{X}(t, \mathbf{x}))$. Assume $\mathbf{X}(t, \mathbf{x})$ follows the ODE (4)

$$\begin{cases} \frac{d}{dt}\mathbf{X}(t, \mathbf{x}) = \mathbf{v}(t, \mathbf{X}(t, \mathbf{x})), & \mathbf{x} \in \Omega, \quad 0 \leq t \leq T, \\ \mathbf{X}(0, \mathbf{x}) = \mathbf{x}, \end{cases}$$

| Model | nb params (pre-trained model) + GP | epochs | nb layers | batch size | lr | nb decay euler |
|---|---|---|---|---|---|---|
| **eight Gaussian** | | | | | | |
| BNAF+GP | (15.4K) + 332 | 2000 | 20 | 1000 | $10^{-2}$ | 5 |
| **two moons** | | | | | | |
| BNAF+GP | (15.4K) + 332 | 1000 | 15 | 1000 | $2 \times 10^{-3}$ | 5 |
| **pinwheel** | | | | | | |
| BNAF+GP | (15.4K) + 332 | 800 | 15 | 1000 | $2 \times 10^{-3}$ | 4 |

Table 3: Parameters used for the training of GP flows on the 2D toy examples.

We recall that the notation $d/dt$ must be understood as deriving the first variable of $\mathbf{v}(t, \mathbf{X}(t, \mathbf{x}))$ while $D/Dt$ represents the derivative in time of the function $f(t) := \mathbf{v}(t, \mathbf{X}(t, \mathbf{x}))$. By deriving the first and second variable with respect to $t$ one has

$$\frac{D}{Dt}\mathbf{v}(t, \mathbf{X}(t, \mathbf{x})) = \frac{d}{dt}\mathbf{v}(t, \mathbf{X}(t, \mathbf{x})) + (\frac{d}{dt}\mathbf{X}(t, \mathbf{x}) \cdot \nabla)\mathbf{v}(t, \mathbf{X}(t, \mathbf{x})).$$

Using $\frac{d}{dt}\mathbf{X}(t, \mathbf{x}) = \mathbf{v}(t, \mathbf{X}(t, \mathbf{x}))$ one finally obtains

$$\frac{D}{Dt}\mathbf{v}(t, \mathbf{X}(t, \mathbf{x})) = \frac{d}{dt}\mathbf{v}(t, \mathbf{X}(t, \mathbf{x})) + (\mathbf{v}(t, \mathbf{x} \cdot \nabla)\mathbf{v}(t, \mathbf{X}(t, \mathbf{x})).$$

## C  Experiment details

We run the experiments on two separate GPUs: a NVIDIA Quadro RTX 8000 and a NVIDIA TITAN X. Our loss is given by the negative log-likelihood (1). The FFJORD model has an inverse function directly available in the code, which is not the case for the BNAF model. Therefore, as explained in the Section 5, the GP flow is trained using $\mathbf{f}^{-1}$ applied on a standard multi-dimensional normal distribution for FFJORD, whereas only the training data are used for BNAF.

**Toy datasets.** We give the parameters used for the 2D toy experiments in Table 3[3]. We consider a training set of 80K samples and a testing set of 20K samples, 15 time steps with a Runge-Kutta 4 discretization and a GP flow with two intermediate layers of size 15. For the Euler penalization we take the exact same parameters with $\lambda = 5 \times 10^{-4}$ which is divided periodically by a factor 2 the number of division is given in table 3.

**dSprites, MNIST and chairs datasets.** The VAE architecture used in the experiments is taken from the github repository of Yann Dubois (Dubois et al., 2019). The parameters used for the GP flows are given in Table 4 and the architectures of CPFlow are given in Table 5. For all test cases we consider a GP flow with 15 time steps, three intermediate layers of 50 parameters for the dSprites dataset and four intermediate layers of 100 parameters for the MNIST and chairs dataset. For the Euler penalization we take an initial parameter $\lambda = 5 \times 10^{-5}$ which is then divided 10 times periodically by a factor 2 during the training.

## D  Additional details on 2d examples

We provide additional details on some toy examples presented in Section 6.1. First we consider the eight Gaussian test case, study the transformation of a uniform mesh by the NF model and compare with the CPFlow architecture. As shown in Figure 7, the mesh transformation when adding the GP flow is getting closer to the CP-Flow one, and the OT cost is roughly the same. Another illustration of the transformation of the source distribution is given at the bottom of Figure 7. We clearly see the added value of GP flow, as the points' configuration gets much closer to an isotropic distribution indicating that we get closer to the Monge map. Note that adding the GP flow does not affect the estimated density nor the test loss.

| Model | # params (pre-trained model) + GP | epochs | batch size | lr |
|---|---|---|---|---|
| **dSprites dataset** | | | | |
| FFJORD+GP | (17.8K) + 10.3K | $\approx 60*$ | 1024 | $5 \times 10^{-4}$ |
| FFJORD+GP+EULER | (17.8K) + 10.3K | 1250 | 1024 | $5 \times 10^{-4}$ |
| **MNIST dataset** | | | | |
| FFJORD+GP | (36.9K) + 40.6K | 1250 | 1024 | $5 \times 10^{-4}$ |
| FFJORD+GP+EULER | (36.9K) + 40.6K | 1250 | 1024 | $5 \times 10^{-4}$ |
| **Chairs dataset** | | | | |
| FFJORD+GP | (36.9K) + 40.6K | 1250 | 1024 | $5 \times 10^{-4}$ |
| FFJORD+GP+EULER | (36.9K) + 40.6K | 1250 | 1024 | $5 \times 10^{-4}$ |

Table 4: Parameters used for the training of GP flows. Each epoch is made of $2.4 \times 10^5$ samples randomly generated from a normal distribution.
* The training is stopped early due to out-of-domain particles.

| Model | nb neurons | nb layers | nb blocks | Total nb params |
|---|---|---|---|---|
| CPFlow (1 block) | 128 | 10 | 1 | 89.7K |
| CPFlow (3 blocks) | 64 | 5 | 3 | 36.6K |

Table 5: Architectures used for CPFlow for the dSprites, MNIST and chairs datasets.

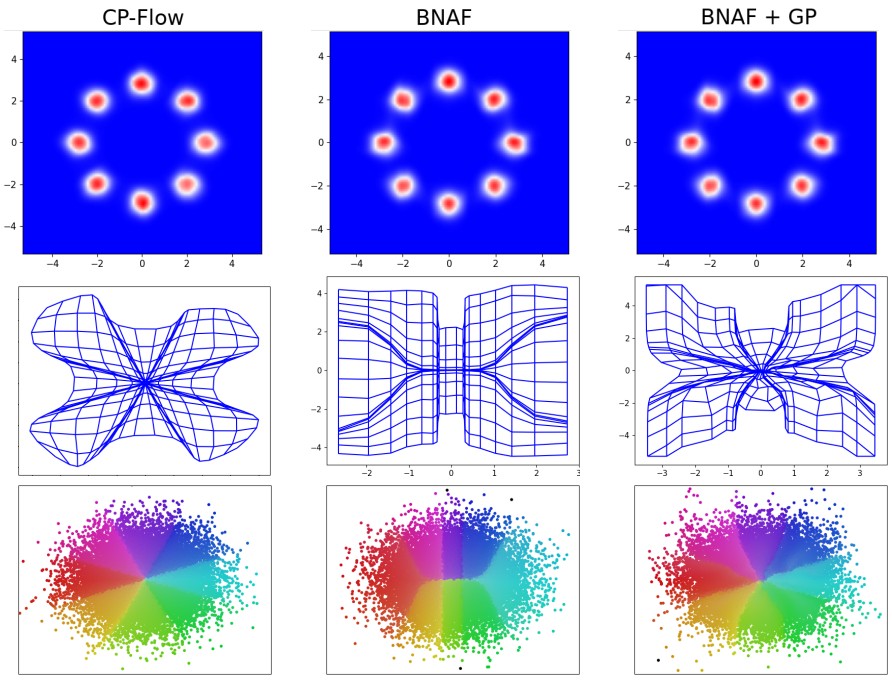

Figure 7: Eight Gaussian test case. Top: density estimation. Middle: deformation of a uniform mesh by the NF model. Bottom: Colored source distribution. From left to right: CP-Flow (OT=2.62, loss=2.86), BNAF (OT=2.88, loss=2.85), BNAF+GP (OT=2.60, loss=2.85).

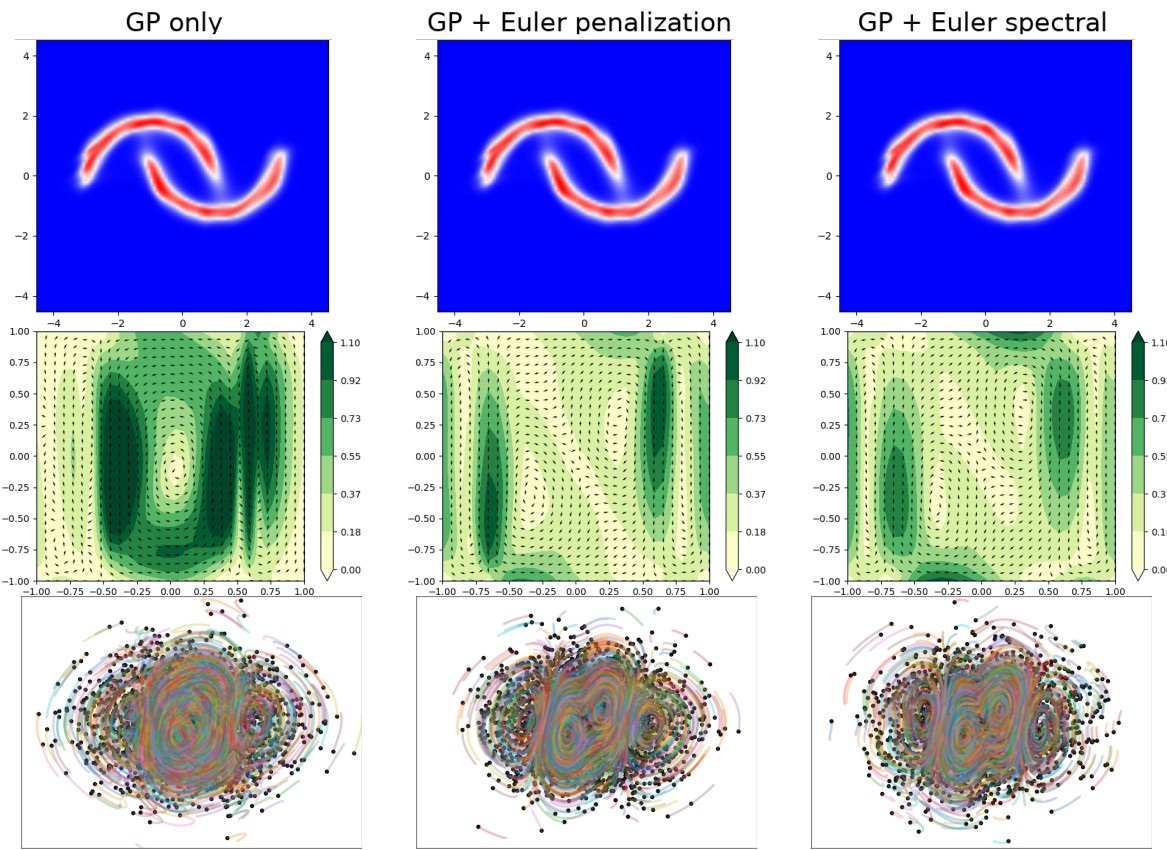

Figure 8: Two moons test case with the BNAF model (initial OT=1.35). Top: target density after the application of the GP flow. Middle: representation of the initial incompressible velocity field $\mathbf{v}_0$ of the GP flow in $(-1, 1)^d$ (where the color map represents the norm of the vector field)
. Bottom: trajectories of the Gaussian particles under the action of the GP flow. From left to right: GP only ($\bar{\mathcal{E}} = 0.28$, OT=1.05), GP with Euler through the penalization procedure (12) ($\bar{\mathcal{E}} = 0.09$, OT=1.04), GP with a spectral method solving directly the Euler equations ($\bar{\mathcal{E}} = 0.08$, OT=1.06). For the last two mentioned the initial velocity fields and the trajectories are very similar and the energy $\bar{\mathcal{E}}$ is lowered showing that the penalization procedure efficiently solve Euler's equations.

**Euler's penalization.** We now turn our attention to the two moons dataset and evaluate how close our penalization procedure is from the true solution of Euler equations. To this end we compare our solution with a more standard method where the initial condition is optimized through a numerical scheme (implemented in Pytorch to be differentiable) which directly solves the Euler equations, in the line of "differentiable physics" approaches (de Avila Belbute-Peres et al., 2018). The baseline numerical scheme is a spectral method, a very efficient and popular numerical method (Canuto et al., 2007). We do not go into details on how to implement such methods since they are restricted in practice to low dimensions and we refer to Canuto et al. (2007); Quarteroni (2009) for a complete presentation. To have a more precise comparison we use 20 layers and a learning rate of $5 \times 10^{-3}$. On Figure 8, we compare three different cases: GP alone, GP with Euler penalization and GP with Euler constrain through a spectral method. As indicated by the OT costs the final positions of particles are very similar for each model. We observe that the trajectories, the initial incompressible velocities and the energy $\bar{\mathcal{E}} = \frac{1}{N} \sum_{i=1}^{N} \int_{0}^{T} \frac{1}{2} \|\mathbf{v}(t, \mathbf{x}_i)\|^2 dt$ are very similar both for the penalization-based procedure and the spectral method which show that we can correctly solve the Euler equations in 2D with our penalization approach (the additional benefit being the generalization to high dimensions). Without the addition of the Euler constraint however, the initial velocity field and the trajectories look very different resulting in a higher energy, and thus a transformation that does not correspond to a geodesic in $\mathrm{SDiff}(\Omega)$.

## E Interpolation examples

---

[3] The 2D test cases can be found for example here https://github.com/rtqichen/ffjord/blob/master/lib/toy_data.py

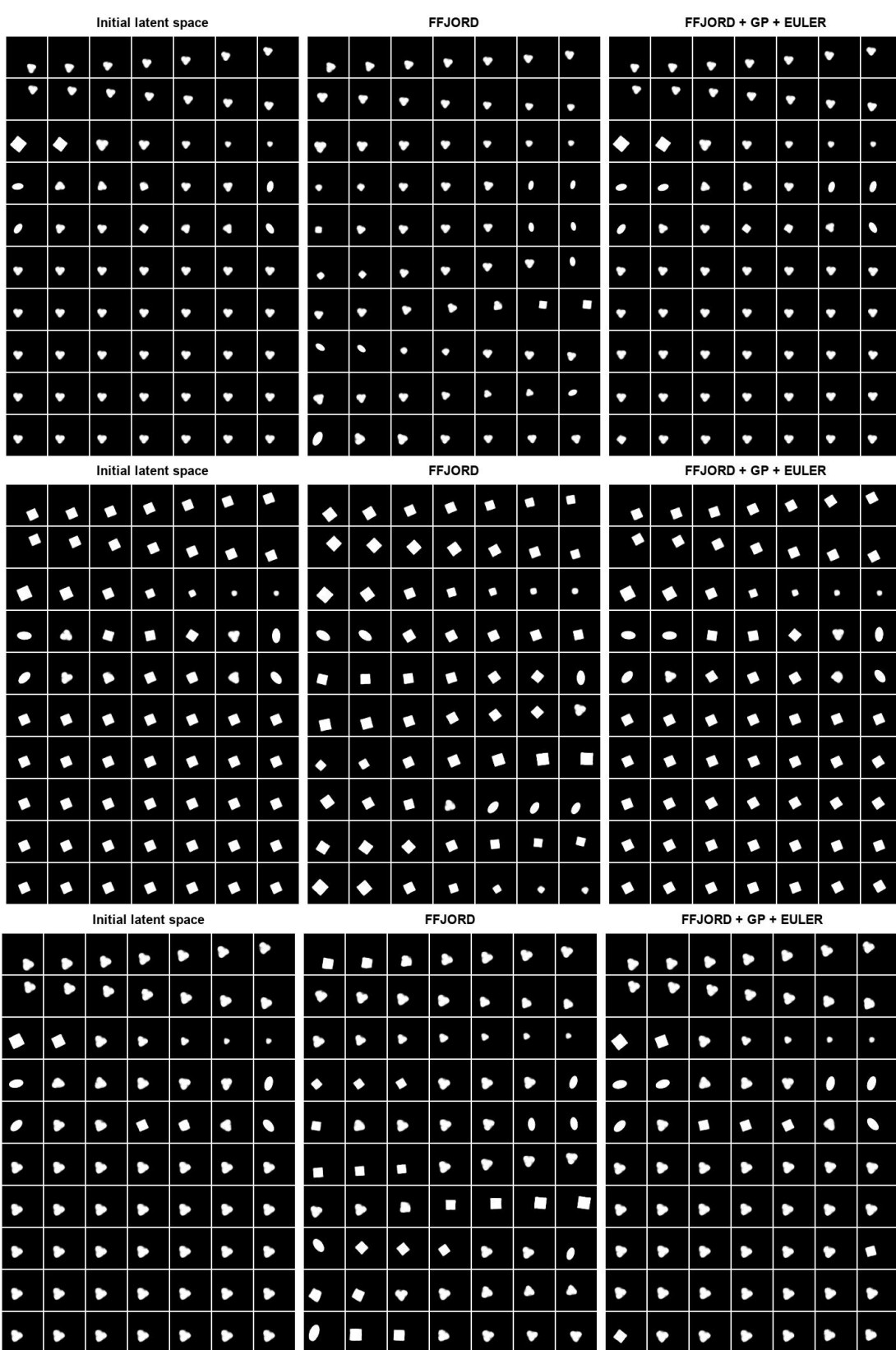

Figure 9: Examples of interpolation for the dSprites dataset where each block correspond to the interpolation of a different data point along the 10 dimensions axis represented by the rows. The dimensions are sorted with respect to their KL divergence in the VAE latent space, so the higher rows carry more information while the last rows should leave the image unchanged.

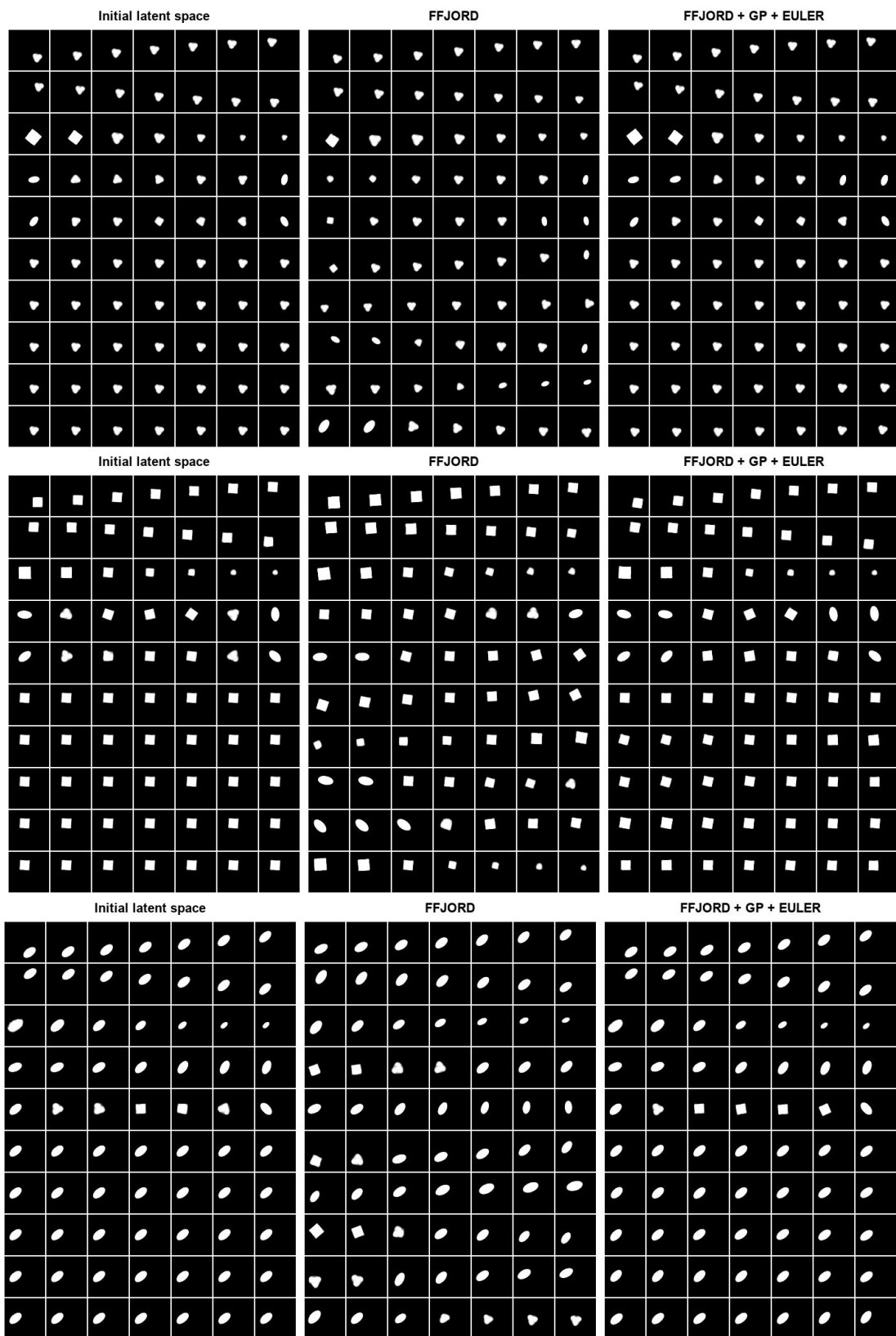

Figure 10: Examples of interpolation for the dSprites dataset where each block correspond to the interpolation of a different data point along the 10 dimensions axis represented by the rows. The dimensions are sorted with respect to their KL divergence in the VAE latent space, so the higher rows carry more information while the last rows should leave the image unchanged.

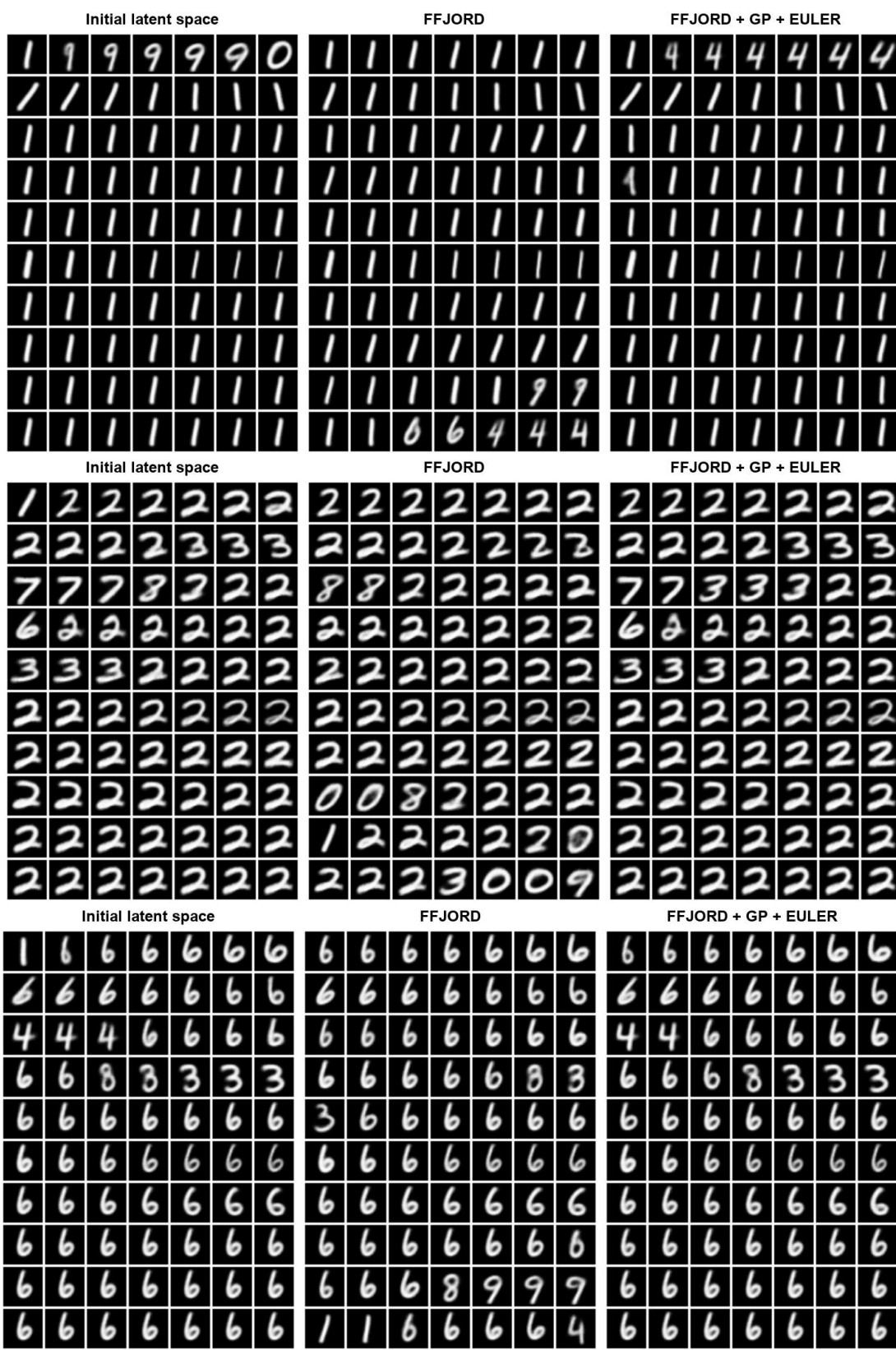

Figure 11: Examples of interpolation for the MNIST dataset where each block correspond to the interpolation of a different data point along the 10 dimensions axis represented by the rows. The dimensions are sorted with respect to their KL divergence in the VAE latent space, so the higher rows carry more information while the last rows should leave the image unchanged.

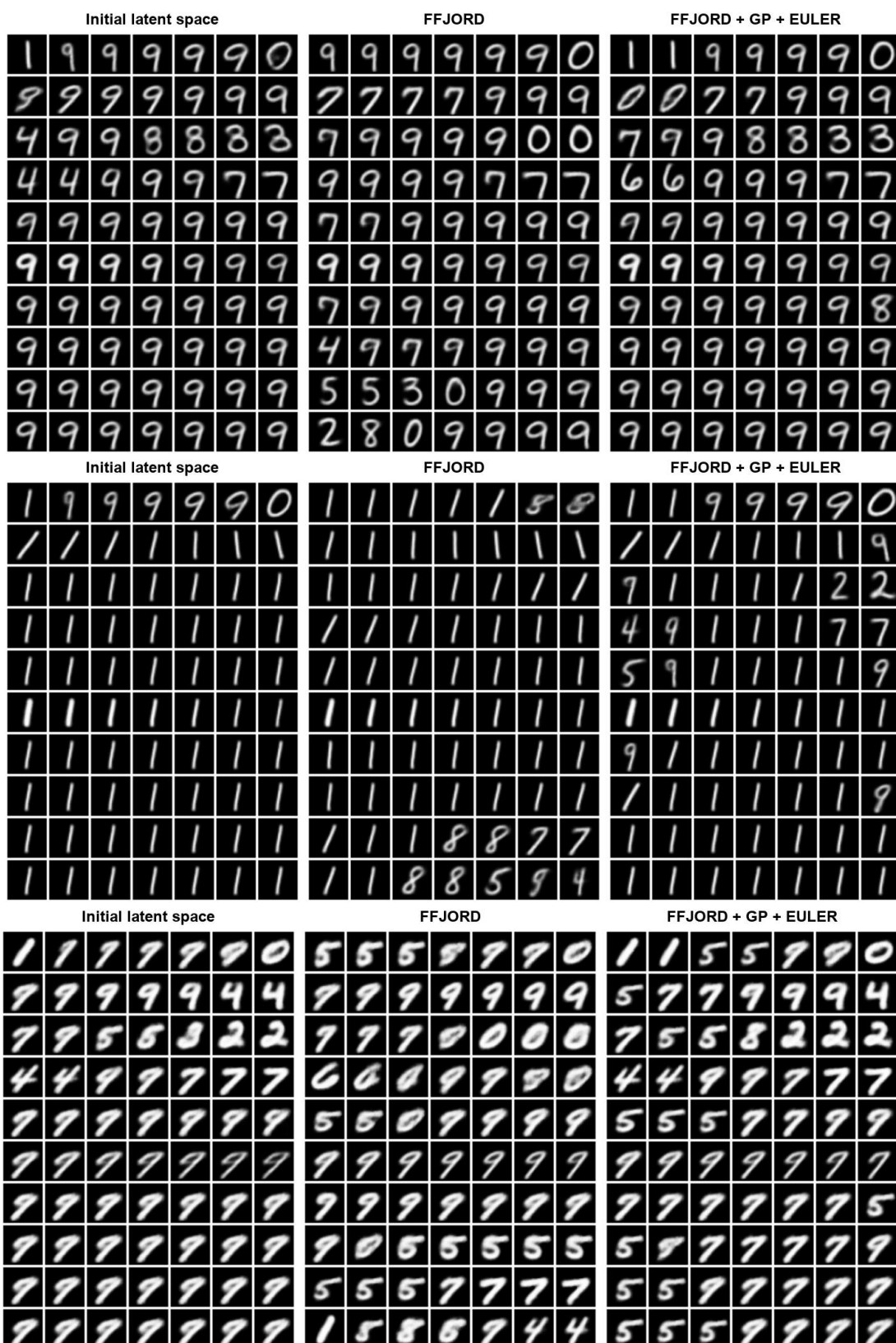

Figure 12: Examples of interpolation for the MNIST dataset where each block correspond to the interpolation of a different data point along the 10 dimensions axis represented by the rows. The dimensions are sorted with respect to their KL divergence in the VAE latent space, so the higher rows carry more information while the last rows should leave the image unchanged.

Figure 13: Examples of interpolation for the chairs dataset where each block correspond to the interpolation of a different data point along the 10 dimensions axis represented by the rows. The dimensions are sorted with respect to their KL divergence in the VAE latent space, so the higher rows carry more information while the last rows should leave the image unchanged.

Figure 14: Examples of interpolation for the chairs dataset where each block correspond to the interpolation of a different data point along the 10 dimensions axis represented by the rows. The dimensions are sorted with respect to their KL divergence in the VAE latent space, so the higher rows carry more information while the last rows should leave the image unchanged.

