# OpenReview forum: "Turning Normalizing Flows into Monge Maps with Geodesic Gaussian Preserving Flows"
_TMLR — Accepted by TMLR_

### Review · Reviewer_XMTg · 2023-02-08

**Summary Of Contributions:**

This paper proposes a method to transform a given trained normalizing flow (NF) mapping a Gaussian to a distribution of interest (or vice-versa, depending on whether the NF or its inverse is considered) in such a way that the updated flow (1) still maps a Gaussian to the same target distribution, and (2) the optimal transport (OT) cost of the updated NF is now optimal (i.e. the updated flow should be a Monge map between the target distribution and the Gaussian, that is, out of all the transformations mapping the Gaussian to its target, the one achieving minimal transportation cost).

In order to do this, the authors leverage Brenier's polar factorization theorem, which states that, under suitable conditions, the Monge map can be written as a composition of two functions: the first one a measure-preserving function $s$ (in the setting of interest, this just means that $s$ maps standard Gaussians to standard Gaussians), and the second one which can be written as the gradient of a convex function. The authors then treat the given pre-trained NF as the gradient of the convex function, and aim to find learn the measure-preserving function $s$ which will minimize the transport cost. The authors then take $s$ as a continuous normalizing flow, given by the solution of an ODE defined by a parameterized vector field $v$. With the help of Arnold's theorem, the authors characterize the vector fields which result in a measure-preserving solution to the ODE, thus obtaining constraints on the vector field $v$. The authors enforce these constraints both through architectural choices and a regularization term in the training objective.

**Audience:**

Yes

**Broader Impact Concerns:**

I have no broader impact concerns.

**Claims And Evidence:**

Yes

**Requested Changes:**

Address the concerns I raised in the weaknesses above, particularly points 4-6.

**Strengths And Weaknesses:**

## Strengths:

1. As pointed out by the authors, OT-based NF work considering Brenier's polar factorization theorem often focuses on the gradient of the convex function and not on the measure preserving function $s$. I agree with the authors that there is value in bringing this point to the community's attention.

2.  I believe the paper is technically sound, although I have some doubts about whether the method is actually solving the OT problem as claimed in the paper, or just reducing (but not minimizing) the transport cost. I will detail my doubt in the weaknesses section of my review.

3. Despite some minor points that I will also outline in the weaknesses section of my review, the paper is well-written and understandable.

## Weaknesses:

4. My biggest issue with the paper lies in its motivation: I am not convinced that the problem being solved is of any practical relevance. While I can understand the motivation behind training NFs with an OT-based loss in the first place (e.g. the loss might be better behaved than maximum-likelihood, and the achieved models somehow improved; or the architectural constraints might not require efficient computation of the log determinant term) as is done is some of the related works mentioned in the paper, I am really struggling to see how improving the transportation cost of an NF -- while not actually changing the learned distribution -- can be used for anything.

5. I believe the issue I raised above in point 4 is reflected in an unconvincing experiments section: (a) Figures 2 and 3 and Table 2 show the proposed method indeed better matches the Monge map than baseline NF models, but no explanation as to why this might be desirable is provided. (b) The comparisons on disentanglement are similarly unilluminating as to why one should care about the proposed method: Table 1 shows that applying the proposed method on disentangled representations obtained with a VAE achieves better disentanglement than applying FFJORD on the same disentangled representations. It is certainly not common practice to train an NF on disentangled representations from a $\beta$-TCVAE, and it is once again unclear why anyone would do so in practice. The experiment feels very forced, trying to showcase benefits where there might be none. I think it should also be pointed out that the proposed method gets better disentanglement than this forced baseline, but does not improve upon the disentanglement from the initial VAE's representations, once again highlighting the issue I have with the paper's motivation. Additionally, I am in general skeptical of results claiming disentanglement [1].

6. I do not believe the current experiments convincingly show that, even if one decides to only care about the OT cost and not about how well the NF maps a Gaussian to its target (an assumption that, as discussed above, I do not agree with), then the proposed method is the way to go. The comparison against CPFlow on Table 2 seems to be the only attempt at establishing this, yet: (a) it is odd to carry out the comparisons on the latent space of a VAE rather than on actual data, which I suspect is due to scaling issues due to the parameterization of the vector field, (b) it is unclear that the comparisons are computationally fair, i.e. is the compute used for the FFJORD+GP+EULER or FFJORD+GP models comparable to that of the CPFlow model? The proposed models essentially use two NFs, the pre-trained FFJORD model, plus the Gaussian preserving GP model, which could easily result in a more flexible model than CPFlow, and thus explain the good empirical performance of the proposed method, and (c) it is also fair to wonder if all the complications introduced by the authors are worth it: one can obtain volume-preserving flows in a much easier way (albeit not continuous normalizing flows), e.g. NICE [2]. Showing that the introduced measure-preserving flows that the authors use outperform this naive choice should also be carried out.

7. The authors claim they seek to find the optimal map in OT cost, yet it is unclear to me if this is actually what their objective does, or if the objective can merely decrease but not actually recover the Monge map. Through Theorem 1, we know that the optimal $g$ should be of the form $\nabla \psi (s(x))$, where $\psi$ is a convex function. Later, the trained NF is plugged in as $\nabla \psi$, so that only $s$ is learned by minimizing OT cost. I do not think that the pre-trained NF needs to match the gradient of a convex function (let alone that of an optimal $\psi$), meaning that the set of functions expressible as applying the pre-trained NF to $s(x)$ need not actually contain $g$ (even if one assumes that NFs can represent any diffeomorphism). Similarly, Lemma 2 shows how one can obtain a vector field satisfying a desired condition, but not that any vector field satisfying the required condition can be written as in Porposition 2, which might also mean that the Monge map is not recoverable. Could the authors please elaborate on these points? Am I missing something, or am I correct that the proposed method will only reduce OT cost but not necessarily recover the Monge map? To be clear, even if I am indeed correct, I would not see this as a fundamental issue, just one requiring to tone down language about what the proposed method achieves.

8. In remark 1, the authors point out that the assumptions to Brenier's theorem are not exactly fullfiled as the considered support of the Gaussian is unbounded. Yet, when constructing Gaussian-preserving maps, the authors map back to the bounded set $(-1, 1)^d$ and construct measure-preserving maps (i.e. "uniform preserving" maps) there. Can we not reinterpret this as trying to find Monge maps between the uniform distribution on $(-1, 1)^d$ (rather than a Gaussian) and the target distribution? In other words, can we not still rely on Brenier's theorem? If not, it should be explained why not, and if yes, this should be explicitly mentioned in the paper.

9. Finally, I point out some typos/minor errors throughout the manuscript:
- citations are incorrectly typed in LaTeX, please use both \citet{} and \citep{}, depending on context
- "A third popular class..." -> "A fourth popular class..."
- "smooth diffeomorphism" is somewhat tautological, as diffeorphisms are $C^1$ by definition, specially if the exact meaning of "smooth" is not specified in the paper
- please introduce notation before using it, e.g. with the pushforward operator
- "In practice the flow must satisfy the change..." -> "In practice the flow satisfies the change..."
- "\mu := g_{#} \nu" in section 1.1 is not how \mu is defined
- "gaussians" -> "Gaussians" (several times throughout the manuscript)
- the change from $f$ to its inverse $g$ halfway through the manuscript is confusing, and a single direction should be picked from beginning to end
- Theorem 1 is not formally stated: (a) a probability space is a triplet of a set, a $\sigma$-algebra, and a probability measure, not a pair as in the theorem, more importantly (b) the assumption that $g$ is "non-degenerate" should be formalized, and (c) the meaning of measure preserving function should be defined before stating Theorem 1
- I believe there is a missing negative sign in the definition of $\beta(x)$ in remark 1, and in the definition of the erf function (which by the way should be bolded for consistency) before proposition 1
- $\Omega$ is introduced as $(-1,1)^d$ halfway through, you should either denote this set as $\Omega$ from proposition 1, or not use it at all
- in the definition of SDiff($\Omega$), the first comma should be a colon
- the definition of $\Omega$ is changed in Lemma 2 from an open to a closed set
- it is not clear what the pressure $p$ is in equation 8
- I believe stronger statements than those made in section 4.2 hold: I believe $w_{t,x} = \nabla p(t,x)$ if and only if $\nabla w_{t,x}$ is symmetric (rather than only one direction), and similarly $M$ is symmetric if and only if $y^\top M z - z^\top M y = 0$ for all $y, z$. I think this is important to highlight, as it means that the used regularizer actually encourages the desired properties
- "comparison.As shown in Figure 3" -> "comparison. As shown in Figure 3"
- the claim that your method "does not constrain the NF architecture to obtain the Monge map" is a bit misleading, as constraints are incorporated in the construction of $v$


[1 ] Challenging Common Assumptions in the Unsupervised Learning of Disentangled Representations, Locatello et al., 2018

[2] NICE: Non-linear Independent Components Estimation, Dinh et al., 2015

---

> ### Author Response · Authors · 2023-02-28
> **Response to reviewer XMTg (1/3)**
>
> Thank you for the detailed review and the constructive comments. Please find our answers below.
>
> ## Summary Of Contributions:
> > the Monge map can be written as a composition of two functions:the first one a measure-preserving function $s$ (...), and the second one which can be written as the gradient of a convex function.
>
> The notations that we used in Theorem 1 were probably a bit confusing and we have corrected them. Actually the normalizing flow $g$ can be any invertible $C^1$ function then there exists a measure preserving function $s$ such that the Monge map denoted $\nabla \psi$ can be written $\nabla \psi = g \circ s$ (see also our answer to Point 7).
>
> ## - Points 4 and 5
> > About motivation and experiments.
>
> Thank you for pointing out these points. The manuscript has been modified to clarify the main motivation of this work, which is the preservation of data structure with OT transformation.
>
> Preserving data structure can be very useful for example when trying to interpolate between data points. Indeed interpolating directly in the data space often leads to poor results mainly because we may interpolate through low density regions. In this case it can be a good idea to map the data density into a Gaussian (with a normalizing flow for example), interpolate in the Gaussian space and then apply the reverse transformation. However for the interpolation to be meaningful we expect the NF architecture to preserve as much of the data structure as possible (for example if we think of the worst case possible where the data points are randomly mapped to the Gaussian the interpolation becomes meaningless).
>
> This was our main idea for the application of GP flows. Unfortunately, while the data structure is easy to show in 2d (since we can visualize it), it is much harder to study in high dimensions. This is why we needed a data space which satisfies two properties:
>
> - The initial data space should be very structured.
> - The interpolations between data points should be easily comparable and in particular we should be able to classify them easily.
>
> A disentangled data space satisfies these two properties and interpolations along the axis are easily comparable since only one variation factor should change. In our case, therefore, the goal was to get as close as possible to the initial interpolations since this showed that even if we transformed the initial density to a Gaussian we could preserve the data structure at least up to some extent. In particular the goal was **not** to improve the initial disentanglement.
>
> Overall, we do not have a strong opinion on disentanglement: we have seen some very enthusiastic papers about it, but indeed the article you mention seems to be more skeptical. However, we do believe that preserving the general data structure may be useful in some cases, and this is the main application that we have in mind for GP flows.
>
> We have modified the introductory paragraph of Section 6.2 to make the goal of this experimental section clearer.
>
> ## - Point 6
>
> > it is odd to carry out the comparisons on the latent space of a VAE rather than on actual data, which I suspect is due to scaling issues due to the parameterization of the vector field
>
> Indeed, as mentioned in the limitations, GP flows do not currently scale up to the image dimension, which is one of the reasons why we made comparison in a VAE latent space. As explained above, the other reason is that we needed a data space with a lot of structure.
>
> ### Regarding the comparison with CPFLows
>
> We understand that this may be misleading as we are putting all the values in one table. However, our goal was not to make a direct comparison with the CP flow architecture, but instead only to answer the question "Are we close to the Monge map?".
>
> While this question is easy to check in 2d, it is much more challenging in high dimensions. In fact, all  the state-of-the-art papers we have in mind of NF + OT do not directly check if they are close to the Monge map, but instead only show some benefits in terms of training or evaluation procedure (in our case we have similarly shown that our OT approach improves data structure preservation).
>
> Let us mention that the high OT costs observed in our experiments with CPFlow could not be fixed in practice by making the training longer (even if it further reduced the loss). Nevertheless in our opinion a direct comparison between GP flow and CP flow would be of limited interest since the comparison will depend on the initial NF chosen for the GP flow (plus GP flows do not scale as well as CP flow for the moment). As mentioned in the perspectives, GP flows could still be useful since there may be cases where CP flow is simply unapplicable when the initial NF architecture is already constrained to satisfy other non OT properties. This is not the case for GP flows which could still be applied whatever the original constraints were.

---

> > ### Author Response · Authors · 2023-02-28
> > **Response to reviewer XMTg (2/3)**
> >
> > In conclusion, the most relevant parameter here in our opinion is that the two losses between FFJORD and CP flow are roughly the same in order to have comparable OT costs. Since we get OT values which are always lower than the CP flow ones with FFJORD+GP for similar losses, we conclude that we are at least as close to the Monge map as CP flow.
> >
> > Note that as pointed out by another reviewer we have considered CPFlow with 3 blocks in our experiments and therefore CPFlow is not guaranteed to converge to the Monge Map (it is guaranteed only with 1 block). We will rerun the experiments with 1 block but from our early results it seems that it is more difficult to make CPFlow converge in this case.
> >
> > We have rewritten the paragraph about CP flow in section 6.2 to make it clear that we are using CP flow only as a baseline to ensure that GP flow is close to the Monge map.
> >
> > ### Additional reference
> >
> > In complement to our answers above, we would like to point out the recent article [1] which was not available at the time of submission. In [1], the authors discuss related motivations to ours. In particular, they discuss the constraints required for ICNN architectures (to which CPFlow belongs) which may be challenging to impose in practice. They also applied other OT costs (not only based on the $L^2$ ground metric) which, as explained in the perspectives of our paper, is also a possibility for GP flows:
> >
> > "(...) Despite their mathematical elegance, fitting OT maps with ICNNs raises many challenges, due notably to the many constraints imposed on $\theta$; the need to approximate the conjugate of $f_\theta$; or the limitation that they only work for the squared Euclidean cost."
> >
> > [1] Ambrosio Luigi and Marco Cuturi, The Monge Gap: A Regularizer to Learn All Transport Maps, 2023. https://arxiv.org/pdf/2302.04953.pdf
> >
> > ### NICE transformation
> >
> > The reason why we do not use the volume preserving transformations given in the NICE paper is that we need the additional property on the domain of the volume preserving transformation which is $s: (-1,1)^d \rightarrow (-1,1)^d$. This does not seem to be possible with the transformations from the NICE paper.
> >
> > ## - Point 7
> >
> > > Through Theorem 1, we know that the optimal $g$ should be of the form $\nabla \psi (s(x))$, where $\psi$ is a convex function ...
> >
> > In Theorem 1, we use the notations from Brenier's work but as we mentioned at the beginning, this is confusing. We should write $g(s(x)) = \nabla \psi (x)$ instead of $g(x) = \nabla \psi (s(x))$. The two notations are equivalent since if $s$ is measure preserving so is $s^{-1}$ but the notation $g(s(x)) = \nabla \psi (x)$ is more consistent with the rest of the paper. Therefore, the normalizing flow $g$ only needs to be a $C^1$ function and a measure preserving function $s$ can be found such that $g(s(x)) = \nabla \psi (x)$ where $\psi$ is a convex function (and $\nabla \psi$ is therefore the Monge map). We have modified the notations in Theorem 1.
> >
> > > Similarly, Lemma 2 shows how one can obtain a vector field satisfying a desired condition, but not that any vector field satisfying the required condition can be written as in Porposition 2, which might also mean that the Monge map is not recoverable.
> >
> > That's a good point, we should indeed highlight it. Indeed, at the moment we do not know if there is a way to prove that every divergence free vector field satisfying the boundary conditions can be written under the form of Lemma 2. Since in practice we observed that we can at least significantly reduce the OT cost with this construction, we decided to leave this investigation for future work.
> >
> > Note that this concerns only the boundary conditions (we still have all the divergence free functions with Proposition 2). Therefore, even if it affects the OT cost, it may have a limited impact since when we are away from the boundaries Proposition 2 gives a universal approximation result of the divergence free functions.
> >
> > We have modified Lemma 2 and added a comment just after to be clearer.
> >
> > ## - Point 8
> >
> > > Can we not reinterpret this as trying to find Monge maps between the uniform distribution on $(-1,1)^d$ (rather than a Gaussian) and the target distribution?
> >
> > Please note that we have the additional change of variable with the function erf which prevents to use directly Brenier's theorem on the uniform distribution. It may indeed be the case that Brenier's theorem could be adapted to this work without any major difficulty but this would require to carefully go through Brenier's proof again which we believe is not the primary focus at the moment.

---

> > > ### Author Response · Authors · 2023-02-28
> > > **Response to reviewer XMTg (3/3)**
> > >
> > > ## - Point 9
> > >
> > > Thank you for bringing to our attention these typos / minor errors we have addressed in the new version of the manuscript.
> > >
> > > > in the definition of the erf function (which by the way should be bolded for consistency) before proposition 1
> > >
> > > We do not bold the erf function just before Proposition 1 as it is the one-dimensional erf function. The bolded one is a vector function where this 1D function is applied component wise.
> > >
> > > > it is not clear what the pressure $p$ is in equation 8
> > >
> > > From a mathematical point of view equation (8) is just a way of saying that $\partial_t v + (v \cdot \nabla) v$ should be equal to the gradient of some scalar function. In addition with the divergence free and boundary conditions this guarantees uniqueness of the vector field $v$. Additionally the pressure field can be interpreted as the Lagrange multiplier of the divergence free constraint for the associated variational formulation of Euler’s equations. We have added a comment on that.
> > >
> > > > the claim that your method "does not constrain the NF architecture to obtain the Monge map" is a bit misleading, as constraints are incorporated in the construction of v
> > >
> > > Our function $v$ is indeed constrained. What we meant is that the initial NF architecture is not constrained and all the training related to the density estimation can be done without any OT-related constraint. The training of our GP flow is separated and can therefore be done whatever the initial NF architecture and / or training procedure was.
> > >
> > > As mentioned before, when the NF architecture has some strong initial constraints, it could be difficult to further constrain it to write the CPFlow formalism for example.

---

> > ### Comment · Reviewer_XMTg · 2023-03-06
> > **Acknowledgement of rebuttal**
> >
> > I thank the authors for their detailed reply and for the manuscript updates. I also appreciate the fact that the authors highlighted in red the manuscript modifications, as I find that openreview's diff functionality to not be too practical.
> >
> > I am actually satisfied with the reply about the motivation as preserving data structure, and agree that the presented experiments on the latent space of VAEs support the claims in the paper. I also agree with the reply about NICE not being as straightforwardly applicable as I thought when I wrote my review, and thus consider the major issues I had with the paper as having been addressed. I will update my review accordingly.

---

### Review · Reviewer_v55J · 2023-02-15

**Summary Of Contributions:**

The paper under consideration proposes a new idea as well as a method capable to improve a pretrained Normalizing Flow model from the perspectives of Optimal Transport theory. The idea is to train an additional model in the latent space of a NF (which is supposed to be Gaussian) which preserves the latent distribution and forms together with the NF the optimal transport map between the latent distribution and data distribution. The proposed approach is built on specifically-designed volume-orientation preserving maps trained with additional Euler regularization, which improves the mass transformation properties of a learned map. The method is shown to be restoring the disentanglement properties of VAEs with disentangled latent space destroyed by prior NF models.


**Audience:**

Yes

**Claims And Evidence:**

Yes

**Requested Changes:**

* The first question concerns the comparison between the proposed approach and CP-Flow model [2]. The CP-Flow is build to be a composition of gradients of convex functions, which is not obligatory to be gradient of a convex function, i.e., OT map. And it is not clear from the paper whether the authors utilize a single building block of CP-Flow (single ICNN model) when reporting OT costs (Table 2 and Figure 3). This question should be clarified.
* Moreover, it is interesting to compare the proposed approach (pre-trained NF + GP flow) with alternative methods (apart from CP-Flow) which solve the OT problem between latent data distribution and Gaussian distribution directly. Of the special interests are OT solvers based on dual formulation of OT combined with ICNNs [4] or utilizing min-max formulation (MM:R solver from [1]). Note, that min-max based solver presents promising results for OT-related problems [5].
* It is not clear from the paper why does the authors utilizes rather difficult Euler equations-based regularization (eqs. (11)) instead of direct penalization of Energy (eq. (9)). This choice should be explained, especially since the (eq. (9)) - based regularization are successfully used [3].
* What do the different shades of green mean in Figure 4?
* It is not clear for unprepared reader why does the property $\text{div} v = 0$ (eq. 5) imply $\text{det} \nabla \phi = 1$. Some links/clarifications should be added.
* Where does pressure $p(t, x)$ come from in the equations (8). Is it an arbitrary scalar function?
* To my point of view, it is not clear, whether the obtained $s_{\theta}$ combined with NF recovers the true OT map between the latent distribution and data distribution. There should be more convincing practical validation, at least in toy 2D scenarios. This concern is directly related to the question raised in the point 1: If the CP-Flows used contain more than one ICNN blocks, then the OT costs reported in Figure 3 could be different from real OT costs. My recommendation is to consider more extensive practical (low-dimensional) validation, given that there are methods which recover true OT mappings mentioned in the point 2.

[1] Korotin et. al., Do Neural Optimal Transport Solvers Work? A Continuous Wasserstein-2 Benchmark https://openreview.net/forum?id=CI0T_3l-n1

[2] Huang et. al. Convex potential flows: Universal
probability distributions with optimal transport and convex optimization. https://arxiv.org/pdf/2012.05942.pdf

[3] Onken et. al., OT-Flow: Fast and Accurate Continuous Normalizing Flows via Optimal Transport https://arxiv.org/pdf/2006.00104.pdf

[4] Makkuva et. al., Optimal transport mapping via input convex neural networks https://proceedings.mlr.press/v119/makkuva20a.html

[5] Korotin et. al., Neural Optimal Transport, https://openreview.net/forum?id=d8CBRlWNkqH


**Strengths And Weaknesses:**

*Strengths:* To the best of my knowledge, the submission is the first work which presents volume-orientation preserving maps in the context of Machine Learning and corresponding applications. In spite of the presented method is somehow complicated, the idea is fresh and worth to be presented in ML community. Secondly, the problem statement (improving NFs by rearranging the latent points) is also new.

*Weaknesses:* At first, the proposed method seems to be rather difficult, both in terms of computational complexity and algorithmic implementation. Additionally, the scalability of the approach is also questionable (and it is mentioned by authors as a limitation). Secondly, the method seems to have rather limited scope. The only practically-sound problem under consideration is related to NFs in latent space of VAEs and the idea is to restore disentanglement properties of original VAE latent space corrupted by NF.  The setup considered in the paper prescribes that at first we train a VAE-based model, then we train NF on the latent space of the obtained VAE, and finally we train the GP flow proposed. However, it seems that the last two steps could be easily merged by considering recent OT solvers (including CP-Flow) mentioned in the points 1-2 of Requested Changes section. It is unclear how the proposed methods performs compared to these conventional ways to solve OT.

---

> ### Author Response · Authors · 2023-02-28
> **Response to reviewer v55J**
>
> Thank you for the careful review and insightful comments. Please find our answers below.
>
> > it is not clear from the paper whether the authors utilize a single building block of CP-Flow (single ICNN model)
>
> This is a very good remark thank you for bringing this to our attention. Indeed by default we chose to follow the original CP flow parameters which were using multiple blocks in their experiments (in the experiments of Table 2 we used 3 blocks). This may explain why the CP flow architecture was not close to the Monge map.
>
> We will rerun the experiments from Table 2 with only 1 block. The early results we obtain for CPFlow with 1 block are the following
>
> |         | dSprites | MNIST | Chairs |
> |---------|----------|-------|--------|
> | OT cost | 5.65     | 26.08 |2.61    |
> | Loss    | -16.89   | 0.942 |6.60    |
>
> For the MNIST and Chairs dataset it seems that CPFlow with 1 block has trouble converging.
>
> - Regarding the dSprites test case CPFlow converges but the OT cost is higher compared to what we obtain with GP flows.
> - For the MNIST dataset the OT cost is too high. As we explain in the paper it may come from a bad generalization of the backward function which is applied to the Gaussian distribution.
> - For the chairs dataset, the OT costs of GP and CP flows are comparable, GP flow obtained an OT cost which is lower but we note that since the losses are not close to each other they may not represent the same distribution.
>
> We will make additional experiments by changing some parameters to see if we can improve the results but from what we observe so far it seems difficult in some cases to make the CPFlow architecture converge with only 1 block.
>
> > Moreover, it is interesting to compare the proposed approach (pre-trained NF + GP flow) with alternative methods (apart from CP-Flow) which solve the OT problem between latent data distribution and Gaussian distribution directly.
>
> This would indeed be interesting unfortunately given the timing constraints of the rebuttal for TMLR we may not have the time to make other comparisons. We hope that the complementary comparisons of CP flow with one block as well as the older ones illustrate that GP flows are able to get close to the Monde map.
>
> Thank you for the reference on the min-max solver, we will look into it.
>
> > It is not clear from the paper why does the authors utilizes rather difficult Euler equations-based regularization (eqs. (11)) instead of direct penalization of Energy (eq. (9)).
>
> In our opinion the main issue is that the two terms of the loss will have opposite objectives. Indeed the global minimum for the energy is $v^2=0$ (that is, the particles do not move) which is obviously not the velocity field which minimizes the OT cost. In fact the global minimum of a loss composed of two terms OT + $v^2$ may not minimize the OT cost. On the contrary if we consider a loss with the two terms OT + euler, the global minimum (theoretically) minimizes the OT cost since we know there exists a solution that both solves the Euler equations and minimizes the OT cost. We have added a comment at the beginning of Section 4.
>
> Note that even in [1] the authors use various other penalization terms in addition to $v^2$. In particular they write their network as the gradient of some scalar function and penalize the HJB equations along the trajectories. These two additional terms are probably required because penalizing $v^2$ alone is not enough to recover the OT map in practice. We believe this is the reason why most papers which focus on OT with NF do not penalize the $v^2$ term directly and prefer to consider other approaches.
>
> [1]  Onken et. al., OT-Flow: Fast and Accurate Continuous Normalizing Flows via Optimal Transport https://arxiv.org/pdf/2006.00104.pdf
>
> > What do the different shades of green mean in Figure 4?
>
> The colormap is used to represent the norm of the vector field. We have added a comment on this in the caption of the Figure.
>
> > It is not clear for unprepared reader why does the property (eq. 5) imply $\det \nabla \phi = 1$ . Some links/clarifications should be added.
>
> We have added a comment.
>
> > Where does pressure come from in the equations (8). Is it an arbitrary scalar function?
>
> Indeed from a mathematical point of view equation (8) is just a way of saying that $\partial_t v + (v \cdot \nabla) v$ should be equal to the gradient of some scalar function. In addition with the divergence free and boundary conditions this guarantee uniqueness of the vector field $v$. Additionally the pressure field can be interpreted as the Lagrange multiplier of the divergence free constraint for the associated variational formulation of Euler's equations. We have added a comment on that.
>
> > There should be more convincing practical validation, at least in toy 2D scenarios.
>
> We have rewritten the 2d part of the experiments. We added other 2d cases as well as new comparisons regarding Euler regularization.

---

> > ### Comment · Reviewer_v55J · 2023-03-21
> > **Response to the authors**
> >
> > I thank the authors for the response. Overall, I am satisfied with the answers to my questions and concerns. Still, I believe, that it is worth to compare with or at least mention the alternative OT-solvers listed in my review. In particular, probably, the min-max solvers such as [5] can provide more accurate, compared to one block of CP-Flow, restoration of ground truth Monge maps. Therefore, the benchmarking with such a method will contribute to practical validation of the proposed approach. On the other hand, I do not insist on the comparison. Overall, from my point of view, the current version of the manuscript looks good.

---

### Review · Reviewer_XNgV · 2023-02-16

**Summary Of Contributions:**

In this paper, the authors propose a novel approach to design Normalizing Flows (NFs) in which the transport mapping satisfies the minimization property of Optimal Transport (OT) (for a squared distance cost). In particular, contrary to many existing approaches the authors do not train a novel NF with a specific architecture or a specific training procedure but instead rely on the Brenier polar factorization theorem [1] which ensures that any function (under mild assumptions) can be decomposed into the gradient of a convex function (the transport plan) and a measure-preserving map denoted $s$. The mapping $s$ is obtained via the minimization of a functional related to the Wasserstein cost.

In this work the authors focus on Gaussian preserving mappings. This framework already encompasses applications in generative modeling and density estimation. In order to define a Gaussian preserving mapping, the authors define a Lebesgue preserving mapping and show how that this equivalently define a Gaussian preserving mapping through a tractable transformation. Flows with velocity fields with zero divergence correspond to Lebesgue preserving mapping. Therefore, the authors parameterize the velocity field of the Lebesgue preserving flow. There exist many solutions to such flows and in order to pick one the authors add a regularization term corresponding to the Partial Differential Equation (PDE) part of the Euler's Equation which defines geodesics in the space of Lebesgue preserving diffeomorphisms.

Combining this regularization term, the original squared cost loss and the special architecture for the zero divergence velocity field the authors then train the Gaussian preserving transformation. They illustrate their approach on toy 2D data and show how OT maps and their approach can improve disentanglement preservation for several datasets (dSprites and MNIST) in the latent space of a VAE, following the setting of [2].

[1] Brenier (1991) -- Polar Factorization and Monotone Rearrangement of Vector-Valued Functions

[2] Dubois (2019) -- Disentangling VAE

**Audience:**

Yes

**Claims And Evidence:**

Yes

**Requested Changes:**

Changes:
* Address the weaknesses highlighted in the previous comment. In particular, more details about the Brenier polar factorization theorem and the motivation of the Euler's equation regularization. I would like to see more ablations/discussions regarding this regularization term.
* The numerics should be improved. It would be nice to showcase how this approach could be made to work in higher dimensional settings. In particular, I think that adding the regularization of the divergence term (instead of fixing it to zero) would be a great addition to the paper.
* I would like to see more discussion regarding the limitation with a Gaussian source.

Minor remarks:
* Why approximating the time-derivative in (12) ? If we have parameterized $v$ with a neural network can we take the gradient w.r.t. the time variable instead?
* Extension to non quadratic  not clear to me
* Abstract: "further constrain" --> "further constrained"
* Looking at (13) would it be possible to jointly (or iteratively) train the flow and the measure preserving mapping $s$? (In that case one would need to add the training loss of the flow $f$ to (13)). That way one could fine-tune the flow to the measure preserving mapping as well.
* End of page 2: "which make it" --> "which makes it"
* Middle of page 2: "require to constraint" --> "require to constrain"
* Page 7: "comparison.As shown in" --> "comparison. As shown in"
* Equation (21) there is a minus sign missing in the Gaussian in the definition of erf.


**Strengths And Weaknesses:**

Strengths:

* I find the paper to be interesting and quite well-written. It should also be of interest for both the Normalizing Flow (NF) and the Optimal Transport (OT) community. Another benefit of this paper is to popularize the Brenier polar factorization theorem [1] which could also be used in other settings where OT is involved.
* I enjoyed the originality and flexibility of the framework. Not training yet another normalizing flow and instead training a "straightening" of the normalizing flow via a Gaussian preserving mapping is an exciting area of research. The fact that the proposed methodology does not depend strongly on the data structure of the problem by only focusing on changes of the "latent" Gaussian distribution (although there is of course some interaction with the normalizing flow $f$ during the training) is definitely an advantage.
* The numerics confirm that the methodology help reducing the OT cost of the normalizing flow which can help with disentanglement.

Weaknesses:
* In my opinion the main current problem of the paper is its numerical experiments which are quite low dimensional (I might have missed something but could not find the dimension of the latent space in the VAE example). A lot of operation seem to be quite costly and to scale badly with the dimension. For example imposing the fact that the vector field is divergence free is done by computing the Jacobian of some $\mathbb{R}^d \to \mathbb{R}^d$ mapping which will require at least $O(d^2)$ operations. This prevents any application in image processing for example (where for small resolution $d=64\times64$ and therefore $d^2 \approx 16M$). It is also not clear how sensitive the loss is to the Euler's equation regularization and how this scales with the dimension, the robustness with respect to this regularization is not studied in depth. The authors seem to be aware of these limitations and propose in the discussion to drop the divergence free requirement and replace that with a penalization.
* On the theoretical perspective I think the link with OT could be better stated. Indeed after looking at the Brenier polar factorization formula in [1] the connection with OT was not immediately clear to me. I could find the uniqueness of s and its expression as a maximiser in Theorem 1.2 in [1]. The expression as a minimizer of the quadratic formula comes from p.382 [1]. From my understanding of Theorem 1 as stated by the author the uniqueness of s was not clear. The connection between Theorem 1, i.e. the Brenier polar factorization formula and OT should be explicitly stated as well. As far as I understand we get that after the rearrangement $g^{-1} \circ s$ also pushes $\nu$ towards $\mu$ (this is stated by the authors) and can be written as the gradient of a convex function. This is the transport plan for the quadratic loss using the uniqueness property of the rearrangement (with $u = \mathrm{Id}$ this time). It would be great to explain and detail a bit more these ideas before diving into the construction of the velocity field.  The uniqueness of $s$ the measure preserving maps also begs the question of the usefulness of the Euler equation regularizer. Indeed by uniqueness of the mapping $s$ we should recover only one mapping (which is the one that makes $g$ a gradient of a convex function). More broadly speaking I think the motivation of the Euler equation regularization is a bit loose. Even in practice it is not clear what are the benefits of these regularizations. Why is it important to have a geodesics in the space of diffeomorphisms?
* While encompassing many settings such as generative modelling and density estimation (at least in the vanilla setting) restriction to Gaussian measures can be harmful if one is trying to solve tasks in which both the distributions are only accessible via datasets (for example in the case of interpolation between two datasets). While this is not the primary goal of the paper, these applications cannot be considered in this setting which is solely available for the Gaussian case. It is not clear (at least to me) how one could extend this approach outside of the Gaussian distribution setting.


[1] Brenier (1991) -- Polar Factorization and Monotone Rearrangement of Vector-Valued Functions

---

> ### Author Response · Authors · 2023-02-28
> **Response to reviewer XNgV (1/2)**
>
> Thank you for the review and your interesting feedback. Please find our answers below.
>
> > I might have missed something but could not find the dimension of the latent space in the VAE example
>
> > In my opinion the main current problem of the paper is its numerical experiments which are quite low dimensional(...) The authors seem to be aware of these limitations and propose in the discussion to drop the divergence free requirement and replace that with a penalization.
>
> The dimension of the latent space is $10$ (mention just before the "dSprites" paragraph).
>
> We agree that the current limitation of the paper is the $d^2$ scaling of the method. This seems to be related to a fundamental property of divergence free functions however which is why we believe future work may consider giving up on the exact implementation of the divergence free functions and penalizing the divergence instead.
>
> >  In particular, more details about the Brenier polar factorization theorem
>
> > From my understanding of Theorem 1 as stated by the author the uniqueness of s was not clear.
>
> Indeed we should highlight the uniqueness of s in Theorem 1.
>
> It is true that Brenier's statement may be a bit confusing since he does not mention the minimisation of the quadratic cost but instead the maximisation of $\int u(x) \cdot s(x) d\mu$ (using Brenier's notations). This is equivalent since
> $$
> \int (u(x) - s(x))^2 d\mu = \int u(x)^2 d\mu - 2 \int u(x) \cdot s(x) d\mu + \int s(x)^2d\mu
> $$
> Both $\int u(x)^2 d\mu$ and $\int s(x)^2d\mu$ are constant with respect to $s$ since $s$ is a measure preserving function. So minimizing the quadratic cost is equivalent to minimize $- \int u(x) \cdot s(x) d\mu$. We added a comment on that in the paper. Note that our notations slightly differ from Brenier's paper to be adapted to the NF framework.
>
> >  The uniqueness of the measure preserving maps also begs the question of the usefulness of the Euler equation regularizer. Indeed by uniqueness of the mapping we should recover only one mapping (which is the one that makes a gradient of a convex function).
>
> > It is also not clear how sensitive the loss is to the Euler's equation regularization and how this scales with the dimension, the robustness with respect to this regularization is not studied in depth.
>
> > Even in practice it is not clear what are the benefits of these regularizations. Why is it important to have a geodesics in the space of diffeomorphisms?
>
> >  I would like to see more ablations/discussions regarding this regularization term.
>
> Indeed, as discussed in Theorem 1 the measure preserving transformation $\mathbf{s}$ is unique but note that in this work we have constructed $\mathbf{s}$ to be the solution of an ODE with divergence free velocity. Therefore even if $\mathbf{s}$ (that is the solution of the ODE at the final time) is unique, there are infinitely many trajectories which reach this final configuration. This is the reason why Euler's regularization is needed to obtain smooth trajectories for our ODE, a property which may be helpful during the training process to minimize the OT cost.
>
> The usefulness of Euler regularization can be seen on the trajectories. We have added some figures on the trajectories (figures 4 and 8) We see that even if the final states (i.e. the solutions at final time) are approximately the same the trajectories obtained with Euler regularization are much smoother.
>
> We believe the effect on the loss of Euler's regularization depends on the considered test case. In some cases, adding Euler regularization does not change anything. For example for our experiments in the VAE latent space it seems that both on the MNIST and chairs datasets we get the same solutions with and without Euler regularization. However on the dSprites dataset we could not minimize completely the OT cost without Euler's regularization. In this case it seems that Euler's regularization plays an important role.
>
> Regarding the scaling with the dimension our approach requires to penalize a term under the form $w^T \nabla f_\theta v$ which can be computed efficiently with a Jacobian-vector product. Penalizing vector Jacobian product has already been done in other contexts and proved to efficiently scale with the dimension [1]. As mentioned before we believe that at the moment, the main limitation for the scaling of the method is due to the exact construction of divergence free vector fields.
>
> To be clearer, we added some comments on Euler regularization and added some figures of the trajectories in the 2d numerical experiments section. We also added a discussion on Euler regularization at the beginning of section 4. Finally, we added a comment on the scaling w.r.t the dimension at the end of Section 4.
>
> [1] Sliced Score Matching: A Scalable Approach to Density and Score Estimation, Song, Yang and Garg, Sahaj and Shi, Jiaxin and Ermon, Stefano, https://arxiv.org/abs/1905.07088.

---

> > ### Author Response · Authors · 2023-02-28
> > **Response to reviewer XNgV (2/2)**
> >
> > > While this is not the primary goal of the paper, these applications cannot be considered in this setting which is solely available for the Gaussian case. It is not clear (at least to me) how one could extend this approach outside of the Gaussian distribution setting.
> >
> > Our approach strongly relies on the fact that the Gaussian distribution is known in order to construct Gaussian preserving functions. Of course, it could be extended to other simple and known distributions without much effort (such as a uniform distribution for example). However the possibility to adapt it to the case of more general unknown distribution probably depends on the application you have in mind.
> >
> > If the goal is to rearrange the points in a unknown distribution $D_1$ according to some fixed cost $C$ it seems possible. For example $C$ could be the quadratic cost of a function which maps $D_1$ to another unknown distribution $D_2$. A possibility then might be to map $D_1$ to a Gaussian, make the rearrangement in Gaussian space with respect to the cost $C$ and then map the points back to $D_1$. In this case we could recover the OT map between $D_1$ and $D_2$ even if none of them is Gaussian. Thank you for mentioning this application: we have added a comment about this in the discussion section.
> >
> > > In particular, I think that adding the regularization of the divergence term (instead of fixing it to zero) would be a great addition to the paper.
> >
> > We agree that the penalization of the divergence term is a very interesting perspective. Although, even if from a theoretical perspective there is no work to do, it would require a lot of numerical efforts to carefully test it, choose the correct parameters (for example since we would add a term to the loss we should weight it appropriately) and run the new experiments. New questions arise as well: for example does Euler's regularization still work if the divergence is not exactly zero?
> >
> > For these reasons we will probably run out of time before testing it and we may leave these investigations for future work.
> >
> > > Why approximating the time-derivative in (12) ? If we have parameterized with a neural network can we take the gradient w.r.t. the time variable instead?
> >
> > Note that the derivative in (12) is the Lagrangian time derivative that is $\frac{D}{Dt}=\partial_t + (v \cdot \nabla)$. Therefore calculating this derivative also requires to take the gradient w.r.t. space of the network. Since we need to penalize the Jacobian of $D/Dt$ for Euler regularization, this would require to compute second order derivatives of the network which is much more costly. For this reason, we choose to approximate D/Dt and compute the jacobian of this approximation.
> >
> > > Extension to non-quadratic not clear to me
> >
> > The extension simply consists of replacing the quadratic cost by another one, and to rearrange the points in the Gaussian according to this new cost. Note that depending on the new chosen cost we could lose some theoretical properties (for example uniqueness). Moreover, the global minimum should be attainable by a continuous rearrangement of the Gaussian points. These theoretical considerations aside, from a numerical point of view, the procedure would be the same.
> >
> > > Looking at (13) would it be possible to jointly (or iteratively) train the flow and the measure preserving mapping ? (In that case one would need to add the training loss of the flow to (13)). That way one could fine-tune the flow to the measure preserving mapping as well.
> >
> > This is indeed a good remark. In our early numerical experiments we tried something related to this, but we ran into some issues to make the flow converge when projecting it onto the space measure preserving functions. Maybe a more careful choice of numerical parameters may fix these issues, but this would require further investigations.
> >
> > > Minor remarks:
> >
> > Thank you for pointing out these typos we have corrected them in the new version.

---

### Decision · Action_Editors · 2023-03-24

**Recommendation:** Accept with minor revision

**Comment:**

The paper has already undergone a revision, which has addressed the reviewers' main concerns, and meets TMLR's criteria. Therefore it is ready for publication, subject to the minor revision described below.

Reviewer v55J, in their final comment, writes:

> Still, I believe, that it is worth to compare with or at least mention the alternative OT-solvers listed in my review. In particular, probably, the min-max solvers such as [5] can provide more accurate, compared to one block of CP-Flow, restoration of ground truth Monge maps.

I would like the authors to mention in the final version of the paper the alternative OT solvers that the reviewer has listed in their review, and comment on them as appropriate (no experimental comparison is necessary).

**Audience:**

The paper is relevant to TMLR's audience, particularly to the normalizing-flow and optimal-transport communities. The paper presents novel methodology, and should be of particular interest to methodology-oriented readers.

**Claims And Evidence:**

After the authors' rebuttal and the paper's revision, all three reviewers are satisfied that the paper's claims are well supported by evidence. Based on the reviewers' assessment, the paper meets TMLR's criteria.

---

> ### Author Response · Authors · 2023-03-30
> **Camera ready revision**
>
> Dear AE and reviewers,
>
> Thank you, we have submitted a camera ready version. Regarding your comment  we have added a paragraph page 12 just before the discussion section where the alternate OT solver are mentioned.
>
> We would like to thank the AE and the reviewers for their time and insightful comments which improve the quality of the paper.
>
> Best,
> The authors